# Ocular Delivery of Therapeutic Proteins: A Review

**DOI:** 10.3390/pharmaceutics15010205

**Published:** 2023-01-06

**Authors:** Divyesh H. Shastri, Ana Catarina Silva, Hugo Almeida

**Affiliations:** 1Department of Pharmaceutics & Pharmaceutical Technology, K.B. Institute of Pharmaceutical Education and Research, Kadi Sarva Vishwavidyalaya, Sarva Vidyalaya Kelavani Mandal, Gandhinagar 382016, India; 2FP-I3ID (Instituto de Investigação, Inovação e Desenvolvimento), FP-BHS (Biomedical and Health Sciences Research Unit), Faculty of Health Sciences, University Fernando Pessoa, 4249-004 Porto, Portugal; 3UCIBIO (Research Unit on Applied Molecular Biosciences), REQUIMTE (Rede de Química e Tecnologia), MEDTECH (Medicines and Healthcare Products), Laboratory of Pharmaceutical Technology, Department of Drug Sciences, Faculty of Pharmacy, University of Porto, 4050-313 Porto, Portugal; 4Associate Laboratory i4HB-Institute for Health and Bioeconomy, Faculty of Pharmacy, University of Porto, 4050-313 Porto, Portugal; 5Mesosystem Investigação & Investimentos by Spinpark, Barco, 4805-017 Guimarães, Portugal

**Keywords:** ocular diseases, sustained ocular delivery, therapeutic proteins, barriers of corneal tissues, nanocarriers, microcarriers, cell-penetrating peptides, hydrogels

## Abstract

Therapeutic proteins, including monoclonal antibodies, single chain variable fragment (ScFv), crystallizable fragment (Fc), and fragment antigen binding (Fab), have accounted for one-third of all drugs on the world market. In particular, these medicines have been widely used in ocular therapies in the treatment of various diseases, such as age-related macular degeneration, corneal neovascularization, diabetic retinopathy, and retinal vein occlusion. However, the formulation of these biomacromolecules is challenging due to their high molecular weight, complex structure, instability, short half-life, enzymatic degradation, and immunogenicity, which leads to the failure of therapies. Various efforts have been made to overcome the ocular barriers, providing effective delivery of therapeutic proteins, such as altering the protein structure or including it in new delivery systems. These strategies are not only cost-effective and beneficial to patients but have also been shown to allow for fewer drug side effects. In this review, we discuss several factors that affect the design of formulations and the delivery of therapeutic proteins to ocular tissues, such as the use of injectable micro/nanocarriers, hydrogels, implants, iontophoresis, cell-based therapy, and combination techniques. In addition, other approaches are briefly discussed, related to the structural modification of these proteins, improving their bioavailability in the posterior segments of the eye without affecting their stability. Future research should be conducted toward the development of more effective, stable, noninvasive, and cost-effective formulations for the ocular delivery of therapeutic proteins. In addition, more insights into preclinical to clinical translation are needed.

## 1. Introduction

Millions of people worldwide are affected by the diabetic retinopathy (DR), a neurodegenerative disorder of retina, which is one of the most common causes of blindness involving other complications, such as retinal vein occlusion (RVO) and corneal neovascularization (CNV) [1].

DR is caused by the damage of blood vessels at the back of the eye and does not show initial symptoms that lead to early DR or nonproliferative DR, where no new blood vessel growth occurs and patients have dilation of pre-existing capillaries, oedema, capillary occlusion, microaneurysms, and intraretinal neo-angiogenesis, leading to tortuous blood vessels formation [2]. Such proliferative vascular changes subsequently turned to severe-type damage to blood vessels and showed growth of fragile leaky blood vessels (neo-angiogenesis) in the retina called proliferative DR that leak a jelly-like substance, filling the center of the vitreous, leading to detachment of the retina from the back of the eye. Patients might observe black spot or floating strings in the vision, blurring or fluctuating vision, dark or empty areas in the vision, hemorrhage in the vitreous, and glaucoma, leading to gradual weakening of the vision [1,2,3]. 

A common cause of vision loss in older people is age-related macular degeneration (AMD), in which patients show degeneration of the retinal pigment epithelial cells and choroidal neovascularization [3]. In dry AMD, the macula thins (atrophic) with age in some patients. In wet AMD, known as neo-vascular AMD, the new vessel growth is the major cause that occurs with abrupt onset of central RVO, leading to capillary occlusion and inducing tissue hypoxia, increasing vascular endothelial growth factor (VEGF) expression and resulting in retinal proliferation of new vessels [3,4,5]. Thus, researchers are investigating new therapies that involve the use of monoclonal antibodies, vascular growth factors, oligonucleotides, genes, and anti-VEGF agents (e.g., ranibizumab, bevacizumab, aflibercept), for the prevention of neo-angiogenesis and stabilization of vascular leakage and, thereby, reducing the oedema.

Several therapeutic proteins have recently been approved on the market for the treatment of ocular diseases (Table 1). Although many of these proteins have low molecular weight (<50 kDa) and short half-life, the physiological and anatomical barriers of the ocular tissues limit their efficacy when administered to the posterior segments of the eye. In addition, the ocular environment makes them unstable and inactive, leading to the failure of the treatment. Among the factors that contribute to this is the presence of proteolytic enzymes, such as trypsin, in the vitreous, which can increase with aging, resulting in degradation of injectable proteins Moreover, various static, dynamic, and metabolic barriers are responsible for short half-lives of therapeutic proteins [6,7].

Anti-VEGF delivery to the posterior segment of the eye by the intravitreal route is very painful, involving the use of a needle to penetrate the globe and release the drug into the vitreous. Moreover, repeated injections are required during the treatment, leading to increased further complications such as cataracts, retinal tears, endophthalmitis, and retinal detachment [7].

Thus, the research focus should be directed at reducing the dosing frequency (e.g., novel prolonged release formulations) and development of novel noninvasive methods or devices for drug administration (e.g., through nonparenteral routes). So far, researchers worldwide have investigated several strategies for the treatment of retinal diseases to minimize the limitations or gap within the current therapies involving therapeutic proteins, reducing patient administration pain while improving compliance. The use of depot formulations of injectable carriers containing drug-loaded micro- or nanoparticles, injectable in situ hydrogels, implants, and cell-based systems are among the most useful approaches to provide safe and sustained ocular delivery of therapeutic proteins [8]. These formulations can improve the ocular drug bioavailability and help reduce the frequency of drug administration, providing an increased drug residence time within the intraocular tissues and improving the treatment efficacy with good patient compliance. In addition, cell-based systems and cell-penetrating peptides (CPPs) are also offering good ocular bioavailability indicated from the phase III clinical trials on an anti-inflammatory peptide conjugated CPP delivery [9]. 

Ideal therapeutic protein ocular delivery systems should provide stable delivery of encapsulated proteins, sustained release, maintenance of effective concentrations at the target tissues, and minimal invasiveness with low systemic exposure. A usual practice is to combine technologies, such as injectable hydrogels containing nano- or microparticles, liposomes, or nanoparticles containing therapeutic proteins coated with bioadhesive polymers [8,9]. Advantages of sustained delivery of therapeutic protein formulations include improved patient compliance, adherence to chronic therapy, and local delivery with fewer side effects and a reduction in dosage and dosing frequency [10].

Currently, great attention is being focused on the development of a more effective noninvasive, sustained drug delivery in the treatment of ocular disorders for the anterior and posterior segments of the eye.

In this review, we discuss the recent approaches for protein delivery to the ocular tissues with a view to increase the patient compliance by increasing bioavailability for longer duration with minor side effects. Different approaches, which include injectable micro/nanocarriers injectable hydrogels, ocular implants, iontophoresis, and periocular injections, are addressed, with a view to improve the ocular bioavailability and provide sustained release to the ocular tissues in posterior segments of the eye of therapeutic proteins.
pharmaceutics-15-00205-t001_Table 1Table 1Molecular characteristics of various antivascular endothelial growth factor (VEGF) antibodies and anti-VEGF agents.MoleculeStructureTypeK_D_ VEGF165 (pM) Equilibrium Dissociation ConstantMolecular Weight M_w_ (kDa)Standard Dose (IVT) (mg/mL)T_1/2_ Vitreous (days)IndicationTargetReferencesBrolucizumab 
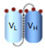
ScFv1.6266/0.0052.94 to 13.4 DR, DME, nAMDVEGF-A[11]Ranibizumab (Lucentis^®^)
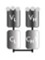
Monoclonal antibody fragment (Fab)46–172480.5/0.0051.4 to 7.19 DR, DME, AMDAll isoforms of VEGF-A[12,13,14,15]Aflibercept
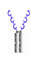
Fc Fusion protein fused with VEFR 1 domain 2 and VEGFR 2 domain 30.4997–1152/0.0051.5 to 5.5 DR, DME, AMDVEGF-A, B and PlGF[16,17]Bevacizumab (Avastin^®^)
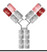
Monoclonal antibody58–11001491.25/0.0054.3 to 11.67DR, DME, AMDAll isoforms of VEGF-A[18,19,20,21,22,23,24,25]Abicipar pegol (Allergan^®^)
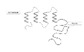
Akyrin repeat protein (recombinant protein) coupled with PEG486 fM342/0.005>13nAMDVEGF-A_165_[26,27]Faricimab
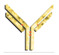
Monoclonal antibody-1506/0.0052.83nAMD, DMEVEGF-A and Angiopoietin-2[28,29]Conbercept (Lumitin^®^, Sichuan)
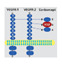
Fc Fusion protein fused with VEFR 1 domain 2 and VEGFR 2 domain 3 & 40.51430.5/0.0054.24AMDVEGF-B and PlGF[30]Pegaptanib (Macugen^®^)
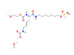
Aptamer (pegylated oligonucleotide200400.3/0.00912DR, DME, nAMDVEGF_165_[31,32,33,34]Abbreviations: DR—Diabetic retinopathy, AMD—age-related macular degeneration, nAMD—neovascularization due to AMD, DME—diabetic macular edema, VEGF—vascular endothelial growth factor, PlGF—placenta growth factor.


## 2. Routes of Ocular Drug Administration

Achieving an efficient ocular bioavailability of different therapeutic proteins remains a challenge due to presence of multiple ocular barriers (Figure 1). Moreover, diseases such as age-related macular degeneration, diabetic retinopathy, and cytomegalovirus (CMV) retinitis require therapeutic proteins to be delivered to the back of the eye. Herein, static barriers (different layers of cornea, sclera, and retina including blood aqueous and blood–retinal barriers), dynamic barriers (choroidal and conjunctival blood flow, lymphatic clearance, and tear dilution), and efflux pumps, in combination, constitute a significant challenge for drug delivery to the posterior segment of the eye [35].

The elimination of therapeutic proteins from the body is similar to the endogenous peptide molecules, i.e., enzymatic cleavage from liver, kidney, blood, and small intestine, although those that show enzymatic resistance can be eliminated via liver or kidney based on their lipophilicity. Only less than 1% of therapeutic proteins with molecular weight >4000 Da show undesirable immune response after administration, which led to the failure of some clinical trials [36]. 

Among the main barriers present in the eye that hinder the ocular delivery of therapeutics are static barriers and dynamic barriers. Static barriers include different layers of cornea, sclera and retina, and blood aqueous barriers (BAB) (Figure 2), while dynamic barriers comprise tear film, choroidal and conjunctival blood flow, and lymphatic clearance, which hinder the movement of drug molecules from the anterior part of the globe to the posterior tissues [36,37]. High selectivity of blood retinal barriers (BRB) limits the movement of topically instilled drugs to the posterior segment. Moreover, systemically administered drugs have to cross the blood ocular barriers, i.e., BAB and BRB, to reach the retina (Figure 2). The use of the systemic route for the delivery for ocular therapeutics has several limitations related to the need of high doses due to systemic metabolism and poor permeability across the BRB. Moreover, exposure to nontargeted organs may cause systemic toxicity and severe adverse effects [37]. 

The topical route is preferred for the delivery of drugs to the anterior chamber of the eye for the treatment of cataract, dry eye, and corneal and conjunctival inflammatory and infectious diseases [37]. The topical ocular delivery route is not commonly used for the delivery of therapeutic proteins for retinal tissues due to the presence of ocular barriers; only <5% of the instilled dose enters through anterior segment to the posterior segment via the tear film and cornea (epithelium, endothelium, and stroma) to the anterior chamber of the eye [8,9,10,38,39]. The extent of absorption of drug molecules from the corneal surface is severely limited by different physiological barriers, such as:

(1) Corneal epithelium that selectively inhibits the diffusion of hydrophilic and high molecular weight molecules such as proteins and peptides through the paracellular route, and it selectively prevents ion transport. Permeability of macromolecules is severely limited by the presence of tight junctions of the cornea and the lipophilic nature of the corneal epithelium.

(2) The endothelium, which is responsible for corneal hydration.

(3) Inner stroma, which presents a hydrophilic nature and inhibits the permeation of more lipophilic molecules [39,40].

These barriers protect the eyes from the entry of toxic entities and pathogenic substances and maintain homeostasis. Moreover, due to the high shear rate, tear turnover, and tear dilution, most (>95%) of the instilled dose is eliminated via the nasolacrimal duct to the gastrointestinal tract, leading to other systemic side effects (Figure 1). The presence of enzymes in the ciliary body digest the drug from the aqueous humor, and the corneal permeability is also limited depending on molecular size, surface charge, and hydrophilicity of drugs [41]. Large and hydrophilic drugs showed poor permeability compared to small and lipophilic peptides from the corneal epithelial tight junction (about 2 nm) [41]. Positively charged molecules can pass easily due to binding with the negatively charged corneal membrane [39]. 

Lipophilic drugs are distributed to corneal tissues via the transcorneal pathway (i.e., cornea > aqueous humor = iris = ciliary body > anterior sclera > lens), while hydrophilic drugs tend to move toward the posterior chamber via the conjunctival–sclera pathway. Large molecular drugs that show poor corneal permeability bypass the corneal epithelium penetration route and undergo noncorneal absorption [42]. Lipophilic peptides with molecular size >700 Da exhibit good membrane permeability [43,44].

### 2.1. Intraocular

Intraocular delivery involves delivery through injection or implants of sterile solutions or devices in the ocular tissues via (1) intravitreal, (2) subretinal, or (3) suprachoroidal routes. 

(a)Intravitreal

The intravitreal route targets drugs to the retina, providing higher drug bioavailability directly into the posterior segment, avoiding several ocular barriers, and eliminating problems associated with systemic toxicities. 

The vitreous has a mesh size of 500 nm that provides a loose barrier and allows diffusion and convection of large and small molecules as well as nanoparticles [45,46,47]. Molecular mobility in the vitreous also depends on the charge of the protein molecules, i.e., neutral and anionic molecules can diffuse more easily compared to cationic ones due to electrostatic interactions with the anionic hyaluronic acid polymer network in the vitreous [48]. Metabolic activity in the RPE determines the bioavailability of protein molecules injected intravitreally due to degradation by the presence of enzymes, i.e., cytochrome P450 and esterases [49]. PEGylation attachment of a high molecular weight hydrophilic moiety to the drug molecules, i.e., polyethylene glycol, either by covalent or noncovalent linkage or encapsulation in the nanoparticles, can dramatically reduce the enzymatic degradation [46].

Clearance observed between the posterior segment and anterior segment after intravitreal injection depends on the size, property, and concentration gradient. Molecules from the posterior segment diffuse to the inner limiting membrane (ILM) and finally reach to retina (Figure 2). The clearance efficiency also depends on penetration efficiency through the tight junction of RPE as the small and lipophilic ones can be transported easily compared to large and hydrophilic proteins [50]. From the retinal layer, the molecules pass through the choroidal blood vessels to the systemic circulation. Molecules that diffuse toward anterior side can be drained away into blood or lymphatic vessels via trabecular meshwork or Schlemm’s canal [50]. 

(b)Subretinal

From subretinal injection, direct administration of molecules to the retina can be possible, so it is the most preferable and efficient route for the delivery of therapeutic proteins with low membrane permeability and of retinal gene therapy [51]. Drugs administered via this route to the inner layer of the retina are cleared via the anterior segment and not through choroidal vessels as RPE tight junctions limit the movement of drugs toward the outer layer of the retina and lead to damage to the RPE and retina (Figure 2) [52].

(c)Suprachoroidal

With the help of microneedles or cannulas, drugs can be administered via this route beneath the sclera into the suprachoroidal space, allowing the drug to be available at the choroidal site. Drug distribution is uneven due to restricted movement from ciliary arteries of the choroid. Moreover, due to high blood flow in the choroidal blood vessels, most of the administered drug is lost to the systemic circulation, which leads to short half-lives. 

Macromolecules such as dextran of molecular weight 40 kDa have an experimental half-life of 3.6 h compared to 5.6 h obtained with a molecular weight of 250 kDa [52]. Bevacizumab (149 kDa) showed even greater half-life (7.9 h) indicating that, apart from molecular weight the charge, flexibility and lipophilicity can also affect the clearance [52]. Through rapid clearance, the particles containing therapeutic proteins form injectable implants with a long retention time that can last up to months. Thus, the suprachoroidal injections of implants containing therapeutics exhibit great scope for effective retinal delivery.

### 2.2. Periocular

It is a less invasive method where drugs are administered directly into the eye via injection into the subconjunctival, subtenon, peribulbar, retrobulbar, and posterior juxtascleral spaces, without any risk of cataract and endophthalmitis. Compared to the topical route, this route provides excellent drug bioavailability by avoiding corneal barriers. Injected drugs reach the posterior segment through the conjunctival sclera, but the bioavailability is much lower (0.1%) than that of the topical route (Figure 2) [53]. Drugs rapidly clearing (80–95%) into systemic circulation through choroidal vessels and multiple barriers between the retina and subconjunctival space leads to poor bioavailability. This route is less invasive and eliminates the drug permeation through sclera. Moreover, in the case of retinal diseases, for drug administration in large volumes, this route is preferred due to the high volume of the injection (100–500 µL) compared with the suprachoroidal route (50–200 µL) [54].

## 3. Ocular Barriers and Approaches to Ocular Administration

### 3.1. Ocular Barriers

Ocular distribution of protein therapeutics to the eye depends on several factors such as membrane permeability, ocular elimination, nontarget binding, and degradation by proteolytic enzymes. Membrane permeability and ocular elimination closely depend on their size, surface charge, and hydrophilicity and lipophilicity [55]. However, complexity of the ocular tissues in deciding parameters for ocular pharmacokinetics is a major obstacle in the designing of an effective delivery system for therapeutic proteins due to the presence of various ocular barriers.

#### 3.1.1. Tissue Conditions

Collagen fibers from the hydrophilic stroma also limit the penetration of therapeutic proteins, which usually takes place via pinocytosis or endocytosis (active transport mechanism) [55,56]. The tight junctions present in the cornea, sclera, and retina significantly prevent the diffusion of hydrophilic large macromolecules [56,57]. The tight junctions in the conjunctival epithelium are usually wider than those in the corneal epithelium but are still unable to provide penetration of large molecules [58,59]. 

The vitreous humor is a highly viscous fluid-like gel composed of 98 to 99% *w*/*v* water content, salts, sugars, a network of collagen-type II fibrils with hyaluronan, glycosaminoglycan, and a wide array of proteins located in the posterior segment of the ocular globe [60]. Drugs administered intravitreally will have direct access to the vitreous cavity and retina and may take several hours to diffuse across the entire vitreous humor. The clearance of macromolecules from the vitreous cavity is very slow due to hindrance by RPE, whereas diffusion from the vitreous to the retina is restricted by ILM [61]. Because several other factors are involved such as initial dose, volume of distribution, and the rate of elimination [62,63], it also depends on size, surface charge, and characteristics of the macromolecules injected [64,65,66,67]. The vitreous can allow the diffusion of small, anionic macromolecules, restricting the bigger size or cationic macromolecules that exhibit nondiffusion kinetics and distribution profile. Molecules can be eliminated through anterior and/or posterior routes [], which is influenced primarily by volume of distribution and elimination half-life [63]. 

A large number of diseases uveitis, cytomegalovirus retinitis, and retinitis and proliferative vitreoretinopathy affect the ocular pharmacokinetics of various topically instilled molecules and their formulations. The diseased conditions produce certain physiological changes in the corneal stroma composed of collagen and water, leading to poor bioavailability of hydrophobic molecules [68]. Fungal keratitis involving chronic inflammation of corneal tissues leads to poor permeation [69]. To solve this problem, drugs are administered with a vehicle/emulsion to avoid evaporation of the limited natural tears in dry-eye patients, as well as the use of the iontophoretic technique to permeate the ionized molecules into ocular tissues.

BRB breakdown as well as choroidal and retinal neovascularization were observed in glaucoma, leading to blindness in a large population. Pharmacokinetic parameters need to be determined in such conditions using animal models to prove efficacy. In one study [70] of measuring the pharmacokinetic parameters, using healthy and diseased animal models, it was observed that the AUC and Cmax were significantly lower in diseased models compared to normal animal models due to BRB breakdown and exposure of drugs to ocular tissues. Therefore, dose calculation needs to be performed to avoid dose-related toxicity. 

#### 3.1.2. Physicochemical Characteristics of Drug Molecules

Various physicochemical parameters of macromolecules such as solubility, hydrophilicity/lipophilicity, molecular weight, size and shape, surface charge, and degree of ionization affect the selection of the route and rate of drug permeation through the cul-de-sac [71]. Small and lipophilic molecules can diffuse and distribute rapidly and largely through RPE, inner limiting membrane (ILM), and outer limiting membranes (OLM), exhibiting efficient distribution to (and even faster elimination from) ocular tissues. Large and lipophilic molecules have poor membrane permeability, showing relatively longer retention time at the site of injection with poor ocular tissue distributions [72,73]. For example, the particles with a size of 200 nm were found to be retained in the retinal tissues for two months after injection [72,73]. The vitreal clearance rate is rapid for smaller particles and can also be observed from their half-lives, i.e., particles of size 50 nm, 200 nm and 2 µm showed half-lives of 5.4 ± 0.8, 8.6 ± 0.7, and 10.1 ± 1.8 days, respectively [74].

Most of the therapeutic proteins have complex structure, large size with molecular weight > 1000 Da, and large hydrogen bonding donor/acceptor groups and show poor membrane permeability across the ocular tissues and barriers [75]. Human retinal tissues prevent the permeation of macromolecules of size > 76 kDa due to inner and outer plexiform layers. Macromolecules greater than 150 kDa cannot reach the inner retinal tissues, while molecules such as brolucizumab (smaller size) can penetrate the retina and choroid tissues more effectively than other anti-VEGF [57,76]. Brolucizumab showed 2.2-fold higher concentrations in the retinal tissues and 1.7-fold higher concentrations in RPE/choroid tissues than ranibizumab in rabbits [77]. These macromolecular proteins, when traversing through the choroid, may wash out through the choriocapillaris, leading to a reduction in therapeutic concentrations, and, due to the large complex molecular structure, may increase the risk of their degradation at the physiological environment of pH and temperature resulting into shorter half-lives. Macromolecules showed half-life in the range of days to a week (Table 1) in the vitreous humor, i.e., bevacizumab had a half-life of 4.32 days with a minimum concentration of 162 μg/mL in the vitreous [78]. Frequent intravitreal injections of ranibizumab 0.3–2.0 mg/eye biweekly or monthly is required to maintain the therapeutic levels as it showed vitreous elimination of 9 days and intrinsic systemic elimination half-life of 2 h, making it noncompliant and often associated with other complications such as cataract, retinal hemorrhage, and detachment and endophthalmitis [79,80]. One comparative study showed brolucizumab clearance from the ocular tissues with a mean terminal half-life of 56.8 ± 7.6 h; ranibizumab took 62 h, and aflibercept was cleared with a half-life of 53 h in the same model [81,82,83]. The rapid clearance is presumed to be due to smaller molecular size and absence of the Fc domain in the case of brolucizumab. Unlike aflibercept, which has full-length antibodies, leading to the conservation mechanism, molecules without the Fc region are more prone to degradation and do not show a cumulative effect even after multiple injections [84]. 

The surface charge being a complex and heterogenous property of amino acid sequence of the therapeutic proteins along with pH of the surroundings are important criteria to be considered. Deamination, isomerization, or post-translational modification of the therapeutic proteins in a particular environment lead to formation of charge variant species in a mixture of therapeutic proteins [85]. Most therapeutic proteins are found to be positively charged at an isoelectric point (pI) of 7–9, leading to charge interactions with other molecules and ocular membranes and showing good penetration compared to negatively charged proteins [85]. Although the undesired entrapment of the polymeric network of the vitreous (negatively charged) should not be ignored, positively charged molecules tend to remain clumped in the vitreous without diffusion, while anionic particles diffuse to the retina [86,87]. The effect of surface charge on the particles was studied on human serum albumin (HSA) and showed that anionic particles of size 114 nm with an overall zeta potential of −33.3 mV can easily diffuse through the vitreal collagen fibrils to the retina within 5 h after injection, while cationic particles of size 175.5 nm with mean zeta potential of +11.7 mV showed aggregation in the vitreous [87]. An inflammatory condition of the vitreous showed accelerated diffusion and clearance of HSA [88].

#### 3.1.3. Viscosity and pH of the Formulation

Most of the protein formulations are available with high and variable viscosity as sustained release of therapeutic proteins for longer duration needs very high quantities to be injected in single-dose administration, which is often associated with high viscosity and difficulty of the syringe to handle the formulation and is not allowed by FDA. A high concentration of therapeutic proteins is very difficult to pass through an 18 mm, 27–30 G needle [89]. Use of viscosity builders required in the formulation of small molecules helps the proteins reach the anterior chamber of the eye in contrast to macromolecules, which helps provide sufficient viscosity to the formulation up to 20 cps, prolong the corneal residence time, enhance the transcorneal absorption into the anterior chamber, and thereby increase bioavailability [90].

pH and osmolarity play a vital role in the ocular therapeutics. For drug delivery to the anterior segment, maximum therapeutic benefits can be achieved when the pH of the formulation matches the lacrimal fluid. The pH of the formulation is a critical parameter that needs to be observed as proteins become denatured and unstable due to irreversible conformational changes at both high and low pH values. Apart from pH, the type and concentration of buffer used can also influence the protein degradation pathways, i.e., deamination, disulfide bond formation/exchange, isomerization, and fragmentation [91,92]. A weak acidic buffer is optimal for the storage of antibodies, i.e., adalimumab (pH 5.2), ranibizumab (pH 5.5), and bevacizumab (pH 6.2), below their isoelectric points (~8.3–8.8) for ocular treatments [93,94]. Though buffers play a crucial role in providing stability and preservation of macromolecules, their use must be carefully considered to avoid associated complications such as immunogenicity and local toxicity [95]. Buffers used also must be within the osmolarity range (280–300 mOsm/kg) to be compatible with ocular tissues as they also impair tonicity. Moreover, hypotonic solutions originate clouding and cause edema of the corneal tissues, while hypertonic solutions desiccate the corneal tissues in the anterior chamber [96]. Therefore, to facilitate protein delivery, proper understanding of the formulation pH and viscosity, selection of buffer system, and use of chemical chaperones are of the utmost importance. This helps to control the behavior and characteristics of the therapeutic proteins and also avoid protein misfolding [97,98]. 

#### 3.1.4. Protein Binding

Protein binding shows less effect on ocular distribution of therapeutic proteins as the level of protein in the eye (0.5–1.5 mg/mL) is significantly less compared to that of plasma (60–80 mg/mL) [99,100]. Vasotide^®^ administered in genetically modified mouse model showed significant reduction in retinal angiogenesis in AMD [101].

Intravitreally administered molecules required to cross the ILM to reach the retina after diffusion through the vitreous body, which contains a high-density extracellular matrix made up of collagen, laminin, and heparin sulphate proteoglycan (composition changes with age), affect the drug permeability [102]. Higher drug penetration was observed with high binding affinity to the extracellular matrix, which led to effective penetration to the ILM, making the drug available to the retina. For example, adeno-associated virus serotype-2 showed excellent transduction to the retina after intravitreal injection due to high heparin sulphate proteoglycan binding affinity, while other serotypes and modified serotypes failed to transfect (low affinity with proteoglycan) [103].

#### 3.1.5. Enzymatic Degradation

Different metabolic pathways also cause the loss of therapeutic activity or inactivation of the macromolecules by protein denaturation, aggregation, precipitations, adsorption, and proteolytic degradation, denaturation by temperature, pH, salt or ionic concentrations, and complexations with enzymes/coenzymes. Enzymatic degradation by proteolytic enzymes depends on concentrations of the enzymes in the vitreous (levels may rise with age and tissue conditions) and on the hydrolytic enzymes and esterases in retina [104,105].

Structural changes in the active form of complex primary, tertiary, or quaternary structures of protein molecules or chemical modification lead to irreversible aggregation and finally inactivation. The main routes of drug administration and fate from ocular tissues are shown schematically in the Figure 1 and Figure 2, respectively. 

Peptides are highly susceptible to enzymatic degradation (proteolytic cleavage) [106]. The proteolytic cleavage and breakdown to small peptides leads to lower half-lives. The drug pharmacokinetic properties and thereby therapeutic efficacy can be achieved by improving bioavailability to the ocular tissues, and that can be achieved by chemical or physical modification of the molecules using various formulation strategies, i.e., coadministration, conjugation of functional moieties, particle formulation, encapsulation into implant or hydrogel, and chemical modification/substitutions. Proteolytic stabilization of macromolecules and membrane permeability can be achieved by a prodrug approach or using biological analogues [55,107,108]. Similarly, lipophilicity or hydrophobicity can also be increased by covalent conjugation with hydrophobic moiety or by noncovalent interactions with any hydrophobic compound. Solubility improvements can also be achieved using a conjugation with cyclodextrin and PEG, eliminating enzymatic degradation [39,40,56,75]. Thus, the pharmacokinetic properties of therapeutic proteins can be optimized, keeping in mind these changes must not affect their biological efficacy. 

Therapeutic proteins need protection against enzymatic attack from the various proteolytic enzymes present in the vitreous such as matrix metalloproteinase and serine/cysteine protease. The level of enzyme concentration in the vitreous changes with the age and disease conditions, so the formulation targeted to the retinal diseases needs to be optimized against such enzymatic attack [109,110]. Use of D-form peptides or peptoid type has been shown to have good enzymatic resistance [111] in addition to chemical modifications at the N and C terminus; for example, C-terminal amidation or N-terminal acetylation will make the peptides more difficult to be recognized and targeted by the enzymatic attack [112,113]. Apart from proteolytic enzymes, certain metabolic enzymes such as cytochrome P450 reductases and lysosomal enzymes are also found in large amounts in the ocular tissues that maintain homeostasis and protect the ocular tissues [114,115,116]. Encapsulation of retinal drugs in a nanoparticulate system or implant matrix can improve the protection against the enzymatic degradation [117], as discussed later in formulation approaches.

### 3.2. Use of Penetration Enhancers 

Different therapeutic approaches have been investigated for the improvement of drug bioavailability and providing sustained drug release to the corneal tissues. Bioavailability improvement to the anterior segment of the eye can be achieved by maximizing corneal absorption and reduction in precorneal drug loss, which can be achieved by using viscosity enhancers, penetration enhancers, and prodrug approaches [80,81,82]. 

The presence of tight junctions in the stratified epithelium allows only ions to be transported across the tissues, offering high resistance to therapeutic proteins; thus, the addition of absorption promotors or penetration enhancers can be more helpful to improve the permeability across the corneal tissues or membrane [53,81,118]. Permeation enhancers alter the integrity of the corneal epithelium, leading to the promotion of the corneal uptake and thus a rate-limiting step in the transport of macromolecules from the corneal tissues to the receptor site [82]. Inclusion of cetylpyridinium chloride [119], lasalocid [120], benzalkonium chloride [76], parabens [118], tween^®^ 20, saponins [64], Brij^®^ 35, Brij^®^ 78, Brij^®^ 98 ethylenediaminetetraacetic acid, bile salts [83], bile acids (such as sodium cholate, sodium taurocholate, sodium glycodeoxycholate, sodium taurodeoxycholate, taurocholic acid, chenodeoxycholic acid, and ursodeoxycholic acid), capric acid, azone, fusidic acid, hexamethylene lauramide, saponins [84], hexamethylene octanamide, and decyl methyl sulfoxide [121] in different formulations has shown a significant enhancement of corneal drug absorption. Moreover, the ability to catalyze the degradation of hyaluronic acid by hyaluronidase is also utilized since it has taken decades to improve the permeability across the ocular tissue barriers [122]. In the vitreous, hyaluronic acid provides a key role in maintaining structural integrity, volume expansion, and viscosity of the vitreous body [123]. Keeping in mind the associated toxicity and irritation, penetration enhancers should be used precisely and carefully.

## 4. Conjugation Approaches

Therapeutic proteins are very potent and offer advantages related to specific mechanisms of action. Despite these benefits, therapeutic proteins have shown several drawbacks, including high molecular weight, short half-lives, instability, and immunogenicity, which must be kept in mind when developing a delivery system. Among the several strategies discussed above, there are few conjugation approaches that are being used to improve the stability and overcome the limitations, called second-generation therapeutic proteins. Change in formulations (i.e., using liposomes and polymeric micro/nanoparticles) or change in the protein itself (i.e., changing the protein structure by attaching covalently some moiety to the protein molecule) are among the few conjugation approaches (Table 2). The covalent conjugation of therapeutic proteins with PEG, hyaluronan, lipid derivatives, and melanin are also better alternatives to modifying the protein moiety itself, as discussed below.

### 4.1. Conjugation with Ligands

Receptor-mediated drug delivery can be used for improvement in drug permeation at the target tissue with a high selectivity, specificity, and efficiency. Delivering therapeutic proteins to the ocular tissues with minimum systemic exposure or minimal toxicity is a great benefit obtained by using receptor/ligand-binding. It also enhances the membrane permeability of most biopharmaceuticals/macromolecules, making the intracellular delivery convenient by enhancing receptor/ligand mediated endocytosis. Few example, transferrin-receptor-specific monoclonal antibody on the surface of transgene-containing liposomes, when introduced into retina-specific drug delivery for CNV treatment, showed expression of transgenes throughout the RPE and multiple areas of ocular tissues [124]. Another study in which an RPE-specific drug delivery platform was used in AMD and vitreoretinopathy as a conjugate (in vitro) observed that CD44-specific RNA aptamer-FITC conjugates were efficiently taken up by the RPE-overexpressing CD44 receptors via a receptor-mediated endocytosis pathway [125].

### 4.2. Conjugation with Lipid Derivatives

Self-assembled peptides modulated via intermolecular (hydrophobic, hydrogen) interactions or conjugation with polymers can be used along with lipidic derivatives [126]. These self-assembled particles can be applied through intravitreal injection using 27–30 gauge needles, which is less invasive compared to those used in implants (25 G) [127,128].

### 4.3. Conjugation with Melanin

A macromolecule derived from tyrosine known as melanin, a polyanionic pigment, showed good affinity with most drugs, exhibiting high retention in ocular tissues [129]. Drugs bound to melanosomes (a form of melanin found in choroid and RPE) found in very high concentrations formed a reservoir and showed slow release over a longer duration, evidence of the good binding affinity of melanin with the drug and the intracellular permeability of drugs. Melanin bound to lipophilic pazopanib and GW771806 showed effective drug retention for several weeks in rat eyes with an ocular half-life of 18 days after a single oral dose of 100 mg/kg [130,131]. Thus, melanin drug binding can be used as a potential carrier for sustained release with extended half-life, facilitating delivery to the posterior segment via noninvasive topical or oral administration. Thus, this can be another potential strategy for delivering fast-eliminating peptides (anti-VEGF proteins), providing good ocular retention and prolonged therapeutic effect.

### 4.4. Conjugation with Hyaluronan

Apart from melanin binding, combining the hyaluronan-binding peptide with anti-VEGF is another approach in which an anti-VEGF protein, when combined with hyaluronan, results in high residence time with a longer sustained release in the vitreous, providing 3–4-fold longer therapeutic effect in corneal neovascularization tested in rabbits and monkeys [131]. These hyaluronan-binding peptides can be used as prodrugs with extended vitreous retention an alternative to frequent intravitreal injections.

### 4.5. Conjugation with Polyethylene Glycol

PEGylated peptides are more preferable for slow release and long retention time in the vitreous. Pegatinib, an anti-VEGF aptamer of RNA combined with high molecular weight PEG (40 kDa), showed prolonged retention due to high molecular size when applied in the treatment of neovascular age-related macular degeneration. Similar results were observed with PEGylated-complement C3 inhibitor-Pegcetacoplan [132]. The vitreous contains a high number of proteolytic enzymes, so ocular delivery of peptides needs protection against such enzymatic degradation for long-term delivery [49,133,134]. PEGylation can shield macromolecules such as peptides and genes and reduces the chances of enzymatic attack [135]. However, no studies have been reported so far for the proteolytic degradation and resistance of drugs in the vitreous. The use of the D-form of peptides instead of the L-form or chemical modification of the C and N terminus of peptides by amidation or acetylation increases the enzymatic resistance, as enzymes do not recognize the peptide [136]. Encapsulation of peptides in nanoparticles or implants can help in minimizing enzymatic degradation during delivery [137].

## 5. Formulation Approaches

Formulation approaches are based on providing sustained drug release to the anterior and posterior segments of the eye, which can be achieved by providing continuous and controlled delivery to the ocular tissues using hydrogels, micro- and nanocarriers, implants, inserts and some modern approaches as CPPs, encapsulated cell technology, iontophoresis, and microneedle formulations [138,139,140,141,142].

Table 2 shows examples of approaches used to improve the bioavailability of ocular therapeutic proteins, and Table 3 describes the most relevant results observed with different formulation approaches used to improve the ocular delivery of therapeutic proteins.

### 5.1. Hydrogels

One of the most promising categories of delivery systems for safe and sustained ocular delivery of therapeutic proteins that is gaining popularity is injectable hydrogels. They are aqueous, highly soft and elastic in nature, and have physicochemical similarities with ocular fluids, being adequate for intraocular use. Moreover, a mild crosslinking of polymers is enough to preserve the biological activity of therapeutic proteins in the hydrogels [143,144,145]. Hydrogels are preferable over other dosage forms for the delivery of proteins, peptides, and antibodies, as the formation of the hydrogel can occur at ambient temperature conditions. These systems can be administered in the vitreous cavity via injection through small gauge needle, as “in situ”-forming aqueous dispersion, which is turned immediately into gel in response to internal or external stimuli mediated by changes in the physiological environment, i.e., pH, temperature, ions, or enzymes [144]. The sol-gel phase transition occurs within seconds to minutes, entrapping and stabilizing the therapeutic proteins in an aqueous polymeric network [146,147,148]. Several in situ gelling polymeric systems prepared with hyaluronic acid, chitosan, poloxamer, HPMC, and polycaprolactone have exhibited safe use as depot systems in the ocular environment. After injection, the hydrogel forms a reservoir, allowing continuous release of loaded protein molecules over time, which restricts their mobility in the polymeric network after the sol-gel phase transition in the ocular tissues. The drug release occurs via different mechanisms from the reservoir, i.e., diffusion-controlled and degradation-controlled [149,150]. The hydrogel’s properties can be modulated for setting the diffusion rate and permeability of entrapped molecules in the hydrogel by different process parameters, such as time, type and degree of crosslinking, and the polymers-to-crosslinker ratio. The level of crosslinking aids in determining the diffusion rate and mechanism of therapeutic proteins from the hydrogels. It also depends on the degree of polymer modification, molecular weight, concentration, density, and polymer architecture [151].

(a)Hydrogels

For decades, tremendous efforts have been made by the formulators to develop biocompatible, biodegradable, fast-gelling hydrogels to overcome challenges such as initial burst release, initial gel viscosity, hydrogel turbidity, crosslinking strategies, sterilization procedures, storage conditions and long-term intraocular stability and safety to facilitate their clinical translation. Many thermosensitive hydrogels become turbid on gelation at body temperature after administration. Moreover, the use of high concentrations in solutions/dispersions for the formulation of long-term depot hydrogels results in formulations too viscous to inject via 22–31 G needles [152,153,154,155].

An injectable hydrogel of PLGA in N-methyl pyrrolidone and sucrose acetate isobutyrate showed sustained release of proteins [147]. Polysaccharide crosslinked hydrogels exhibited sustained release of bevacizumab for three days with a low initial burst, while thermosensitive hydrogels exhibited sustained release of bevacizumab for 18 days [152]. Intravitreal administration of bevacizumab in situ gel of hyaluronic acid-vinyl sulfone and dextran-thiol (HA-VS/Dex-SH) showed controlled release of bevacizumab when tested using rabbit eye model [146,156,157]. In situ hydrogel of bevacizumab administered in a single injection intravitreally in a rabbit eye showed therapeutic concentration (>50 ng/mL) for up to six months with specific binding to VEGF measured using a specific quantification technique, i.e., ELISA assay, at different time intervals [157,158]. A hydrogel formulation with bevacizumab prepared with methoxy-poly(ethylene glycol)-block-poly(lactic-co-glycolic) acid crosslinked with 2,2-bis(2-oxazoline) showed sustained release up to one month using in vivo rabbit model without any cytotoxicity [159].

Another study using a thermosensitive hydrogel of triblock copolymer of poly(2-ethyl-2-oxazoline)-b-poly(e-caprolactone)-b-poly(2-ethyl-2-oxazoline) containing bevacizumab showed good biocompatibility even two months after intravitreal injection, exhibiting sustained release properties [160]. A silk hydrogel containing anti-VEGF therapeutics formulated using silk fibroin as the vehicle for delivery and bevacizumab-loaded hydrogel formulations showed sustained release for three months both in vitro and in vivo in Dutch-belted rabbit eyes when injected intravitreally [161].

A light-activated polycaprolactone dimethacrylate and hydroxyethyl methacrylate in situ hydrogel showed stable and sustained release of bevacizumab for up to four months [162].

A biodegradable thermosensitive poly(N-isopropyl)acrylamide hydrogel showed sustained release of insulin without retinal damage or any inflammatory reactions seven days after subconjunctival injection of the hydrogel [163]. Another study of subconjunctival injection of a biodegradable thermosensitive hydrogel prepared with triblock copolymer of PLGA and PEG containing ovalbumin protein showed 14 days of concentration of the drug in the sclera, choroid, and retina [164]. Bevacizumab released from poly(ethylene glycol)-poly-(serinol-hexamethylene urethane) thermal hydrogel after intravitreal injection in a rabbit eye was observed with sustained release up to nine weeks, which was 4- to 5-fold longer than that observed with free protein injections (2 weeks). The rheological studies conducted using phosphate-buffered saline exhibited a phase transition at 32 °C with maximum elastic modulus at 37 °C [165,166,167].

(b)Combined Hydrogel Systems

Today, nanoparticles have been combined with hydrogels to form a hybrid system for the controlled delivery of therapeutics, especially for localized application and to increase the therapeutic efficacy. Tremendous research has been carried out to show the efficacy of injectable nanocarriers for ocular delivery of therapeutics along with hydrogel systems. A thermo-gelling hydrogel for administration to the eye has been prepared by Cho et al. using thermosensitive hexanoyl glycol chitosan [168]. A nano formulation containing thermosensitive penta-block gelling copolymer for ocular delivery of therapeutic proteins was cited as the platform technology for the ocular delivery of therapeutic proteins via intravitreal injection providing continuous zero-order release without any side effects or potential toxicity associated with targeted ocular tissues [169]. An injectable thermosensitive poly(N-isopropyl acryl amide) hydrogel containing PLGA nanoparticles of protein (ranibizumab/aflibercept) showed an initial burst release followed by sustained release of ranibizumab (0.153 g/day) and aflibercept (0.065 g/day) for up to 196 days [170].

One comparative study of nanoparticles with a nanoparticulate thermoreversible hydrogel containing PLGA-PEG-PLGA polymers showed 1.53 ± 11.1 % reduction in VEGF production in a human RPE cell line in plain drug nanoparticles in comparison with 43.5 ± 3.9% observed with nanoparticulate hydrogel formulation [171]. About 12 weeks of successful long-term in vitro release was observed when encapsulated macromolecules in nanoparticles dispersed in a thermosensitive hydrogel [169]. Overall, the injectable in situ gelling depot system has emerged as a novel and attractive tool for sustained and stable protein delivery to the segment of the eye for as little as a few weeks up to as long as several months.

### 5.2. Particulate Carrier Systems

#### 5.2.1. Microcarriers

The most preferable and useful strategy to provide slow, sustained, and prolonged release of drugs is the use of injectable colloidal particulate scaffolds.

Microparticles are micron-sized carriers that carry active drug molecules and are usually suspended in a liquid media. Several polymeric microspheres are being developed for ocular delivery of therapeutic proteins, and few have reached early stages of clinical trials. The use of biodegradable or biocompatible polymers in the formulation of microparticles, i.e., poly(lactic-co-glycolic acid) (PLGA) [172], polyanhydrides [173], and cyclodextrins [174], eliminates the problems of generating toxic products. These nontoxic products generated can be eliminated easily from the systemic circulation and, thus, cleared safely from the ocular environment [172]. Nonetheless, it is too early to say that microspheres are providing sustained release of macromolecules, maintaining therapeutic levels for longer durations of up to a week or months in ocular tissues. The release of macromolecules from the microparticles is closely associated with the structural properties, surface charge, porosity, size and shape, degradation rate, entrapment efficiency, and molar ratio of the polymers used and the diffusion rate of macromolecules [175]. Additionally, the method of preparation requires caution to protect the structural integrity of protein molecules and preserve their biological activity [176,177].

PLGA microspheres containing pegatinib showed sustained release of up to 20 days when injected via the transscleral route and up to several weeks from intravitreal route [178,179]. Sustained release of ranibizumab (0.153 μg/day) and aflibercept (0.065 μg/day) for 196 days was observed after initial burst release of 22.2 ± 2.1 and 13.10 ± 0.5 μg, respectively, when suspended in poly(N-isopropylacrylamide) injectable thermosensitive hydrogel [180]. Moreover, various techniques have been used to control the degradation and burst release of macromolecules from the microspheres. Using hydrophobic ion-pairing complexation, biocompatible block copolymers that can sustain the drug release for up to three months using stimuli sensitive hydrogel (pH, temperature, enzyme, light, ultrasound, and multiresponsive)-based formulations for ocular delivery of therapeutics are the examples discussed in the final section of this review [181,182]. PLGA microspheres containing interferon-alpha (IFN-α) provide sustained release and antiproliferative efficacy [183,184]. PLGA microspheres with anti-VEGF aptamer EYE001 has been tested in humans and showed sustained release over a period of 20 days [178]. PEG-bevacizumab conjugate, bevacizumab encapsulated with PLGA, and free bevacizumab when studied for reduction in experimentally induced choroidal neovascularization (CNV) showed no significant difference between all three formulations [185]. Sustained release of polylactic acid microparticles with a diameter of 7.6 μm, encapsulating TG-0054 (a hydrophilic drug intended for neovascularization and related diseases) for up to 3 months, was observed in rabbit model. After 3 months, the drug level in the choroid-RPE, retina, and vitreous was similar to that after one month [185]. In another case, the intravitreal injection of the plain injection of the same drug showed low intraocular levels after one month, with no detectable levels after three months [186]. PLGA is a well-known choice for the preparation of microparticles but offers several drawbacks, including poor protein stability, ineffective loading, and fast release profile [22]. A “system-within-system” matrix has been developed to transport ranibizumab to the vitreous for the treatment of age-related macular degeneration [187]. Chitosan-based microparticles in PLGA originated the highest ranibizumab-loading percentage and release when examined along with ranibizumab nanoparticles. Additionally, chitosan-tripolyphosphate-hyaluronic acid microparticles demonstrated antiangiogenic activity owing to hyaluronic acid, which was an advantageous property but one that was regrettably offset by the quick disintegration of the matching PLGA microparticles [188].

#### 5.2.2. Nanocarriers

##### Polymeric Nanoparticles

Polymeric nanocarriers are particles with dimensions between 10 nm and 1000 nm loaded with drug molecules dissolved, entrapped, encapsulated, or adsorbed in natural or synthetic polymers. According to the distribution of the drug molecules in the polymeric matrix, these nanocarriers are categorized as nanospheres (drug molecules are homogeneously dispersed or dissolved in the polymer matrix) and nanocapsules (drug molecules are in the core, which is covered by a polymeric shell) [138]. Injectable nanocarriers have potential for ocular delivery because they provide increased stability to the encapsulated molecules, good ocular residence time, and adherence, leading to excellent bioavailability in the ocular tissues [189,190,191].

Among the different routes for ocular delivery of therapeutic proteins via nanocarriers are topical, periocular, suprachoroidal, and intravitreal routes. As shown in Table 1, injectable polymeric nanoparticles play a significant role in delivering therapeutic proteins via the intravitreal route. More effective and long-term inhibition of corneal neovascularization was observed with intravitreally injected nanoparticles, as compared to other peptides [192,193,194]. The efficient corneal neovascularization inhibition may be due to enhanced pharmacokinetic properties, including prolonged retention time, formation of nanosized depots intravitreally, and avoidance of enzymatic degradation [192,193].

The polymers used provided several benefits such as chemical stability, biocompatibility, tunable degradability, and flexibility in designing the formulation. The polymeric nanoparticles exhibited light-scattering properties, which may cause clouding of the vitreous, loss of bioactivity, and low stability of therapeutic proteins, and need proper attention to the selection of the polymer matrix and nanoencapsulation technique for their delivery through the ocular route [194].

The parameter that determines the stability, particle size, surface charge, or zeta potential of the nanoparticles is the composition of the nanoparticles, i.e., lipidic or polymeric. In addition, this parameter does not affect only at the intravitreal level but also topically and subretinally. Moreover, the use of ligands is important before formulating injectable nanocarriers for the delivery of therapeutic proteins to the ocular tissues (see Figure 3).

A study related to ocular tissue distribution of nanoparticles with different surface modifications (positive and negative charges and PEGylated) using ex vivo bovine vitreous was carried out and showed that particles sized greater than 1000 nm can be used for sustained release of retinal drugs due to the slower clearance from the vitreous [45]. Moreover, high and hindered movements of negatively charged and PEGylated particles and marginal mobility of positively charged nanoparticles were observed. Positively charged particles of size around 200 nm showed marginal mobility when compared to negatively or PEGylated particles of size 500–1000 nm. The mobility of positively charged particles was also affected by the presence of negatively charged hyaluronic acid in the vitreous [195].

Surface charge of the nanoparticles needs to be considered for ocular penetration of therapeutics as higher diffusion in the vitreous was observed with negatively charged human serum albumin compared with positively charged [196]. Thus, negatively charged polymeric nanocarriers are more useful in delivering cationic therapeutic proteins as observed with the delivery of IgG containing gold nanoparticles to the retinal pigment epithelium and photoreceptor cells through subretinal injection [197].

Montmorillonite-chitosan nanoparticles loaded with Betaxolol developed for the treatment of glaucoma showed a positive surface charge of 29 ± 0.18 mV and mean size of 460 ± 0.6 nm and provided strong contact with the negatively charged mucin layer of the corneal membrane. Thus, it was concluded that the developed nanoparticles could provide drug release for longer duration with improved bioavailability [198].

Having the ability to provide sustained drug release, low cytotoxicity, and fewer side effects, PLGA, one of the most commonly used materials among the preferred synthetic or natural biodegradable polymers, has been studied extensively for ocular delivery of therapeutics [199]. For instance, PLGA nanoparticles loaded with dexamethasone after intravitreal injection in rabbits showed 50 days of sustained release with constant drug levels for more than 30 days with a mean concentration of 3.85 mg/L [200]. Based on the areas under the curve (AUC), the bioavailability of dexamethasone in the experimental group was found to be significantly (4.96, 4.15, and 6.35 times) higher than the control group injected with free dexamethasone solution [200].

Both hydrophobic and hydrophilic molecules can be loaded in PLGA nanoparticles [201]. For example, bovine serum albumin, a hydrophilic serum protein, was encapsulated with high efficiency using PLGA nanoparticles [202,203]; lysozyme and human pigment epithelium-derived factor beta 1 separately encapsulated in PLGA nanoparticles provided sustained release across 30 days [204]; in vitro sustained release of the antiangiogenic pigment epithelium-derived factor from PLGA nanoparticles was obtained over 40 days with 70% release within 10 days [205]. The vitreous concentration of bevacizumab was above 500 ng/mL for up to 8 weeks after intravitreal injection of PLGA nanoparticles in rabbits [206]. In the treatment of wet AMD using nano or microspheres of poly(ethylene glycol-b-poly(DL-lactic acid) as the delivery vehicle loaded with bevacizumab, a sustained release of up to 90 days was observed. Moreover, the drug/polymer ratio can be changed to control the drug release rate, and thus, the release rate and bioavailability can be improved as needed [207]. In the treatment of neovascular diseases, nanoparticle-mediated pathway control and expression of natural antiangiogenic factors were found to have significant therapeutic potential in which a proteolytic fragment plasminogen Kringle 5 was found to exhibit sustained angiogenic impact with reduced CNV regions and vascular leakage for two weeks, as observed in CNV models [208,209]. VLN, a low-density lipoprotein receptor extracellular domain, encapsulated in PLGA nanoparticles, exhibited excellent expression of VLN, both in cultured cells and the retina for up to 4 weeks [210]. Another study carried out for the treatment of AMD using anti-inflammatory and antiangiogenic drugs to retinal pigment epithelium showed an antiangiogenic effect for up to three weeks [211]. The PLGA nanoparticles were easily and effectively internalized by ARPE-19 cells via folate receptor-mediated endocytosis, forming a depot and thus providing the sustained effect by downregulating VEGF and upregulating pigment epithelium-derived factor [212]. Aflibercept-loaded PLGA nanoparticles showed 75% drug release in one week, and it has been concluded that it can be more patient-compliant compared to frequent intravitreal injection of plain aflibercept [213]. It was observed that the PLGA microparticles and polylactic acid nanoparticles showed sustained release of bevacizumab (of up to two months) intravitreally in rat model as compared to the effect (of up to two weeks) observed with plain bevacizumab solution when injected intravitreally in rat model [214]. Similarly, PLGA microparticles containing ranibizumab entrapped in chitosan-based nanoparticles for ocular delivery of ranibizumab showed quantifiable release of up to 12 days [215].

##### Micelles

A subclass of amphiphilic nanocarriers that self-assemble in an aqueous environment, producing supramolecular structures in the size range of 10 to 1000 nm known as micelles, has been studied extensively for delivery of small molecules to the ocular tissues [216]. They offer several advantages such as controlled release, reduced toxicity, high drug-loading capacity, and enhanced stability, with highly changeable surfaces, providing high patient compliance [216]. Micelles are prepared using different polymers such as Pluronic F127/F68, N-isopropyl acrylamide, polyhydroxy-ethyl-aspartamide, methoxy-poly(ethylene glycol)-poly(e-caprolactone), and poly(butylene oxide) [196]. Polymeric micelles provide excellent ocular residence time due to their mucoadhesive properties and, having nanosized range, are an excellent system for drugs with poor permeability. Thus, ocular delivery of therapeutics using polymeric micelles showed significant improvement in antiangiogenic therapy for retinal and choroidal vascular diseases and diabetic retinopathy. Anti-Flt1 peptide of GNQWFl, an antagonist for vascular endothelial growth factor (VEGFR1 or Flt1) receptor, (inhibits VEGFR1-mediated cell migration and tube formation), was chemically conjugated to tetra-n-butyl-ammonium modified hyaluronate via amide bond formation in dimethyl sulfoxide using benzotriazolel-1-yloxy-tris(dimethyl amino)phosphonium hexafluorophosphate successfully. It self-assembled itself to form micelle-like nanoparticles in an aqueous solution, showing significant reduction in retinal vascular permeability and deformation of the retinal vascular structure in diabetic retinopathy and effectively inhibiting the CNV in laser-induced CNV in rat model, and it increased the mean residence time of the macromolecule by more than two weeks [207]. Further encapsulation of this conjugated antagonist (anti-Flt1 peptide-HA conjugate) to genistein (tyrosine-specific protein kinase inhibitor) when used as combination therapy in corneal neovascularization showed sustained delivery of more than 24 h with an excellent inhibitory effect both on CNV and vascular hyper permeability [216].

##### Dendrimers

Dendrimers are branched nanosized polymeric carriers, which are layered like liposomes and used extensively for the delivery of ocular therapeutics. Compared to linear polymeric particles, they exhibit high concentrations of payloads of therapeutic proteins. They are monodispersed, having a tree-like architecture based on the polymers, i.e., polyamidoamiones, polyamides, poly (aryl ethers), polyesters, and carbohydrates [217]. The pattern of release, absorption, and elimination of the therapeutics are dependent on the surface charge and molecular weight of the dendrimers, i.e., absorption is at its maximum with cationic charge, and rapid clearance is observed with high molecular weight (>40 kDa) [218]. When using several copies of therapeutics, dendrimers are very useful in controlling the functionality by providing numerous functional groups, which hasten the stimuli responsive ability of dendrimers and help in targeting the connected components and providing binding strength of the ligand to the receptors [218]. A polyamidoamiones hydrogel containing a dendrimer was prepared by crosslinking stimuli-responsive acrylate groups using PEG–acrylate chains activated by UV light and was found to be highly mucoadhesive and nontoxic to the corneal epithelial cells, with high cellular uptake and enhanced corneal bovine transport [217]. Similarly, the polyamidoamiones/PLGA nanoparticulate dendrimer hydrogel was also found to be nontoxic, highly effective, and able to provide the sustained release of the drug on the ocular surface of a rabbit eye [219]. Use of proper conjugation techniques to design different dendrimer conjugates can help in understanding the mechanism of protein adsorption on the surface of dendrimers, also known as “protein corona”, for the use as carriers in ocular delivery of therapeutics [218].

##### Lipid-Based Nanocarriers

Among the injectable colloidal drug delivery systems, the lipid-based nanocarriers are the more interesting and emerging systems also known as “nano-safe carrier systems” for delivering drugs to the ocular tissues due to their excellent biocompatibility, biodegradability, and ability to improve the bioavailability and thereby therapeutic efficacy.

(a)Solid Lipid Nanoparticles (SLNs) and Nanostructured Lipid Carriers (NLCs)

Solid lipid nanoparticles, nanostructured lipid carriers, and liposomes are promising approaches for the safe, sustained, and targeted delivery of therapeutic proteins in many locals, including the ocular tissues. These nanocarriers provide nontoxic, stable, controlled, scalable, and targeted delivery of therapeutic proteins [220]. For example, a sustained release and new synthesis of cytokines in corneal tissues with long-term anti-inflammatory effects can be achieved from p-IL10 containing solid lipid nanoparticles [221]. Recently, a lactoferrin-based nanostructured lipid carrier was evaluated for its stability, cytotoxicity, entrapment efficiency, loading capacity, ocular surface retention, surface charge, and morphology [222]. The results showed nanostructured lipid carriers with an average size around 119.45 ± 11.44 nm, a PDI value of 0.151 ± 0.045, and a surface charge of 17.50 ± 2.53 mV [208]. Moreover, regulated release of lactoferrin, high entrapment efficiency, and lipid content (up to 75%) can be achieved [222]. Nonetheless, more research work is required for understanding the use of solid lipid nanoparticles and nanostructured lipid carriers for the ocular delivery of therapeutics.

(b)Niosomes

There is another self-assembling, nonionic carrier system of lipid-based nanocarriers capable of encapsulating both lipophilic and hydrophilic molecules in their bilayered structured nanovesicles known as niosomes. They release the drug independent of pH, resulting in enhanced ocular bioavailability. Though, similar to liposomes, they are biodegradable, biocompatible, nontoxic, nonimmunogenic, and have good chemical stability, the efficacy of the niosomes as carriers for protein delivery to ocular tissues is still under investigation [223]. Discosomes, another modified version of niosomes, are an excellent strategy for ocular delivery. They are prepared using nonionic surfactants, with a size range of 12–16 nm, and prevent systemic drainage and, thus, improve the ocular residence time; however, they are nonbiodegradable and nonbiocompatible in nature [223]. Persistent protein expression after transfection of pDNA containing niosomes intravitreally for at least one month after injection was found to provide protection against enzymatic digestion and broad surface transfection in inner layer of retina with no cytotoxicity [224].

(c)Liposomes

Liposomes are very small nano- or microsized vesicles containing one or more concentric amphiphilic lipid bilayers and are nontoxic, biodegradable, and biocompatible [225]. The surface charge of the liposomes is very important as positively charged liposomes can make intimate contact with the corneal (negatively charged) and conjunctival surfaces compared to neutral or negatively charged liposomes [226,227]. Bevacizumab-containing liposomes were prepared from PC-PS (cholesterol) to be 100 nm in size using the dehydration–rehydration technique and were coated with annexin successfully for intravitreal administration [226]. The highest transfection effect was observed in low quantities of plasmid DNA in liposomes, with the peak level being reached within 3 days after intravitreal injection of pDNA containing liposomes [228]. In another study, sustained release in the vitreous and retina–choroid from intravitreal injections of liposomal oligonucleotides showed a protective effect against enzymatic degradation [229]. Significant improvement in mean residence time of bevacizumab after intravitreal injection of liposomes [230] and sustained release of bevacizumab from the liposomes were observed for up to 42 days after intravitreal injection [221]. Significantly reduced inflammation with prolonged protection of peptides of up to 14 days in vivo after intravitreal injection of vasoactive intestinal peptide-containing liposomes was also observed [225].

### 5.3. Microbubbles Technology

A new ocular delivery technology using the microbubble, a stimuli-responsive intelligent polymeric carrier system easily converted to nanoscale microbubble vesicles in the presence of stimuli such as pH, temperature, and magnetic field, was found to deliver the therapeutic proteins effectively to the anterior as well as posterior segments of the eye [231].

### 5.4. Nanofibers and Amphiphiles

Self-assembling peptide nanofibers and peptide amphiphiles are also under investigation for use as carriers for ocular delivery of therapeutics. After subconjunctival injection, nanofibers containing LPPR peptide bind to VEGFR1 and NRP1 and significantly inhibit endothelial cell proliferation and cell migration; in addition, abnormal capillary synthesis showed a significant reduction (81.3%) in corneal neovascularization in 14 days in rat as compared to bevacizumab (51.2%), justifying its efficiency to cure angiogenesis-related diseases [232,233,234,235].

### 5.5. Nanowafers

A tiny, transparent circular disc-type or rectangular membrane-shaped nanoreservoir type known as a nanowafer is applied easily with the fingertips and is synthesized using different polymers (e.g., polyvinyl pyrrolidone, carboxymethyl cellulose, hydroxypropyl-methylcellulose, and polyvinyl alcohol) and provides sustained release for a few hours to several days as it remains on the ocular surface for a long period of time [236]. Axitinib nanowafers were tested in ocular burn-induced murine model and observed for inhibitory effects on CNV in mouse model. The results were compared with twice-daily axitinib eye drops (0.1% *w*/*v*) and showed that corneal treatment with nanowafers significantly reduced the proliferation of limbal blood vessels as compared with the corneal treatment with the conventional axitinib eye drops. From the reverse transcriptase polymerase chain reaction study, it was found that the Axi-nanowafer is more effective in downregulating the drug target proteins, i.e., vascular endothelial growth factor (A, R1 and R2), PDGFR-A, and bFGF, compared to that of the conventional eye drop treatment. Thus, once-a-day axitinib nanowafers are more effective than twice-a-day conventional eye drops [236].

### 5.6. Cell-Penetrating Peptides

With excellent membrane modulating ability, cell-penetrating peptides are widely used as penetration enhancers to overcome ocular junction barriers and provide effective drug delivery to ocular tissues. These compounds consist of natural and synthetic amino acid sequences, which facilitate the delivery of peptides, proteins, and genes to intracellular ocular tissues [237,238]. Involving noninvasive or minimally invasive treatments capable of crossing biological membranes, the use of cell-penetrating peptides for ocular delivery has gained increasing attention these days.

Quick translocation of the conjugated protein molecules through the cell membranes into mammalian cells using CPPs as nanocarriers is possible through energy-dependent pinocytosis/endocytosis/direct translocation, which make this an effective strategy for the ocular delivery of therapeutic proteins [238]. CPPs can be integrated within the dosage form, providing sustained release by preventing degradation and increasing the drug residence time.

Intravitreal injection of bevacizumab linked to CPPs (5(6)-carboxyfluorescein-RRRRRR-COOH) with anti-VEGF, topical bevacizumab-CPP with anti-VEGF (twice daily), or dexamethasone gavage (every day) for 10 days revealed a significant reduction in corneal neovascularization areas in all mice for all with in vivo and ex vivo rabbit model [239].

Low cytotoxicity and enhanced permeability of CPPs were reported by Liu et al. [240], who studied the uptake, permeability, and toxicity of various CPPs on human epithelial cells including transactivating transcriptional activator (TAT), penetratin, poly (arginine), low molecular weight protamine, and poly (serine). All the CPPs showed efficient membrane permeability of topically delivered drugs to the posterior segments [240]. In another study, researchers found that the penetratin showed the most efficient distribution of peptides in both anterior and posterior segments of eye as compared to all other CPPs, indicated from significantly higher concentrations of penetratin-conjugated polyarginine (R8) in corneal epithelium and the retina for up to six hours [241]. Topical administration of bevacizumab/ranibizumab conjugated with polyarginine (R6) in a rat eye showed effective concentration of a drug in the retina/vitreous and aqueous humor, which indicated that CPP-conjugated macromolecules showed significantly higher penetration through corneal epithelium and RPE cells, providing effective therapeutic treatment in CNV mouse model [242,243].

Fibroblast growth factor (FGF) administered topically after conjugation with TAT showed effective concentration in the retina for up to 8 h [244]. Similarly, endostatin, an antiangiogenetic protein in conjugation with TAT (22 kDa), when administered topically in mouse eyes, showed significant protein expression resulting in inhibition of CNV [245]. PLGA nanoparticles loaded with macromolecules with TAT linked in the surface showed an excellent efficient delivery in the posterior segment after topical administration [244].

Poor serum stability and toxicity from the CPPs of nonhuman origin were observed in a study performed with CPPs of human and nonhuman origin to overcome immunogenicity [238]. Proteolytic enzymes metabolize noncovalent constraints rapidly, while covalent constraints are proteolyzed by crosslinking with disulfides and amides [246]. Thus, the stability can be improved by increasing the binding affinity of peptides with antibodies using different crosslinks, i.e., lactam, triazole, or thioether group at the helix of peptides [246,247]. The confirmational stabilization of long-chain helix using macrocyclic bridging features or mutagenesis can also be utilized to improve the hydrophobicity and enhanced binding of both covalent and noncovalent constraints [248].

Efficient retinal delivery was achieved for apoliporotein-A1 when fused with penetratin and phospholipids containing high-density lipoprotein particles [249]. Significant therapeutic effect was observed in AMD murine model when pazopanib (an antiangiogenetic therapeutic) contained high-density lipoproteins microparticles along with penetratin [249,250,251].

### 5.7. Encapsulated Cell Technology

Encapsulated cell technology can become the alternative to intravitreal therapy, which uses the expression of protein molecules by providing continuous local production of proteins. Encapsulated cell technology uses permeable materials that allow the diffusion of nutrients, therapeutic factors, and waste products out of cells, thus protecting the cells from digestion by the host immune response. Prolonged release of vascular endothelial growth factor receptor protein (NT-503) and CNTF from implants encapsulating genetically altered human retinal pigment epithelial cells was observed without any retinal degeneration [252,253,254].

A study on long-term cell therapy on mouse eyes using genetically engineered microencapsulated ARPE-19 cells in alginate developed to produce complement receptor-2 fragment (CR2-fH) showed reduced corneal neovascularization and lesion size intravitreally and showed the presence of CR2-fH in the retinal pigment epithelial/choroid of treated mice with systemic expression of fusion proteins without producing any immune response [255,256].

### 5.8. Iontophoresis

Another noninvasive technique to transfer ionized molecules through biological membranes to the ocular tissues using a low electric current is iontophoresis, where the drugs can be transported across the membrane by migration or electro-osmotic processes. Dexamethasone phosphate [257], methylprednisolone [258], carboplatin [259], and methotrexate [260] showed successful delivery through the ocular iontophoretic technique via ocular tissues except therapeutic proteins. Using proper design of devices and probes, one can use iontophoretic technique via the transcorneal, corneoscleral, or transscleral route [261]. As the sclera has more surface area (17 cm^2^) than the cornea (1.3 cm^2^) and is more hydrated, presents fewer cells, and is more permeable to macromolecules having high molecular weight, the transscleral route is the preferred route for ocular delivery of macromolecules to the posterior segment. Moreover, the transscleral route is simple, nonintrusive, has a wide application, reduces the risk of toxicity, and is well tolerated by the patients [262].

The device used for the iontophoresis is flexible and placed under the eyelid to deliver ions through a small area of the eyeball, avoiding tissue damage [263]. Devices such as OcuPhore^TM^ release the drug into the retina and choroid using an applicator, dispersive electrode, and a dose control for the transscleral iontophoresis; Visulex^TM^ is used for the transscleral transport of ionized molecules such as dexamethasone and antisense oligodeoxynucleotides [264,265,266].

### 5.9. Ocular Microneedles

Ocular microneedles are popular delivery techniques exhibiting passive delivery of molecules via arrays of solid microneedles coated with drug formulations that dissolve a few minutes after insertion [267,268]. This technique is used as routine in clinic and is gaining popularity among formulators for having the potential to deliver ocular therapeutics, overcoming the transport barriers of epithelial tissues, eliminating clearance by conjunctival mechanisms, and minimizing retinal damage. For example, sunitinib malate containing microneedle pens showed suppression of corneal neovascularization [269].

### 5.10. Injectable Implants

With the aim of achieving sustained release in the vitreous and providing a long-term therapeutic effect from the polymeric network, intraocular implants are gaining popularity as drug delivery systems. Though it is an intrusive procedure, implants have shown several benefits such as bypassing the blood retinal barrier, avoiding burst release, reduction of the dose, and delivering therapeutics with a constant rate directly at the ocular site [230]. Nonbiodegradable and biodegradable implants are available in the market (see Table 3 and Figure 4) and can have a tunable delivery rate by changing the type and composition of the polymers or the delivery form, i.e., solid, semisolid, or a particulate-based system [270,271,272]. The mechanism of drug release from implants showed three phases, i.e., an early burst, a middle diffusive phase, and a final burst. Different polymers such as polylactic acid, polyglycolic acid, and polylactic-coglycolic acid are being used to prepare implants. Among the advantages of using implants are the no need to perform repeated injections intravitreally, increased half-life, reduced peak plasma level, and improved patient compliance. Nonetheless, the use of this type of device requires surgery or invasive implantation, which sometimes exerts certain ocular side effects. Incorporation of PEG400 and different block copolymers such as PLGA improves the burst release and provides prolonged release of therapeutic proteins [270,271]. Biodegradable implants offer several benefits, such as high payload, prolonged drug release, and minimal burst release, but a minor surgical procedure involves a skilled professional, leading to high treatment cost [273].

FDA-approved biodegradable implants, namely, Ozurdex^®^ based on PLGA and Iluvien^®^ based on polyvinyl alcohol/fluocinolone acetonide in a polyamide tube, used for macular edema and noninfectious posterior uveitis showed sustained release of dexamethasone (0.7 mg) for up to 6 months in the vitreal cavity [274,275]. The short half-life (~3 h) of the corticosteroids leads to faster elimination from the vitreous, which can easily be administered by formulating into implants [276]. So far, no ocular implants have been approved for the delivery of ocular therapeutic proteins, but preclinical studies with human recombinant tissue plasminogen activator showed a release rate of 0.5µg/day in the vitreous for 14 days [277,278,279,280,281].

Nonbiodegradable implants provide more accurate, zero-order, and have longer release rates compared to biodegradable ones, but they require a surgical procedure for their removal, which involves associated risks.

An osmotic implant inserted into the subcutaneous region connected to the sclera using a brain infusion kit delivered IgG for up to 28 days [282]. Another nonbiodegradable implant containing a ranibizumab port delivery system from Genentech showed extended release in the vitreous [283].
pharmaceutics-15-00205-t003_Table 3Table 3Examples of the most relevant results obtained with different formulation approaches used to improve the ocular delivery of therapeutic proteins.Delivery SystemMaterialMoleculeRemarksReferencesMicroparticlesPLGAAnti-VEGF aptamer EYE001In vitro drug sustained release up to 20 days.[178]MicroparticlesPLA nanoparticles in porous PLGA microparticlesBevacizumabIn vivo sustained release after intravitreal injection in rats.[214]MicroparticlesPLGABevacizumabIn vitro drug sustained release for up to 91 days from the microparticles.[284]MicroparticlesPLGABevacizumabIn vivo sustained release after intravitreal injection to rabbit. [285]MicroparticlesSilicon dioxide BevacizumabIn vitro sustained release for up to 165 days. From the porous silicon dioxide microparticles.[286]Microparticles/NanoparticlesPLGA-albuminBevacizumabIn vivo and ex vivo rabbit vitreous injection showed sustained release for up to 165 days from the developed PLGA-albumin microparticles (~197 nm).[287]NanoparticlesHSA-PEGApatinibReduced leakage in vascular tissues with significant inhibition of hyperpermeability in streptozotocin-induced diabetic mice after intravitreal injection of apatinib-HAS-PEG nanoparticles.[288]NanoparticlesAlbuminated PLGABevacizumabSustained release with antibody protection obtained with its stability intravitreally for about 8 weeks with vitreous concentration maintained above 500 ng/mL from the coumarin-6-loaded albuminated-PLGA-NPs.[289]NanoparticlesCS-PLGABevacizumabSustained and effective delivery of bevacizumab to posterior ocular tissues after subconjunctival administration and more reduction in VEGF level in retina than the topical or intravitreal administration.[290]NanoparticlesCS-HA, Zinc sulphateBevacizumabSustained release for up to two months with reduced CNV from CS-loaded bevacizumab nanoparticles containing implants, when administered intravitreally.[291]NanoparticlesPLGAConnexin43mimetic peptideImproved light sensitivity and suppression of inflammatory areas showing high concentration in ganglion cell layer and choroid within half an hour after intravitreal injection. [292]NanoparticlesPLGABevacizumabEnhanced antiangiogenic effect and reduced toxicity of tissues after intravitreal injection in vivo[293]NanoparticlesALBUMINConnexin43 mimetic peptideSignificant enhancement in protection against degradation and high retention in vitro with expression of CD44 cells in both retina and choroid.[294]NanoparticlesCHITOSANBevacizumabVEGF expression inhibition after intravitreal injection. [295]NanoparticlesPLA/PLA-PEOC16Y PeptideSustained release and prolonged effect on suppression of CNV from NPs due to significant permeation to targeted tissues and reduced toxicity as compared to simple peptide solution after intravitreal injection.[296]Nanoparticles12-7NH-12, DOPE, DPPCCy5-DNAConcentration of drug found within 4 h postinjection intravitreally from nanoparticles located in NFL.[297]NanoparticlesChitosanpDNAEffective transfection in INL, IPL, and RGC layers after intravitreal injection.[298]NanoparticlesPLGApDNA, shRNAGFP expression effect is persistent with significant reduction in CNV, lesion thickness, and retinal damage for 4 weeks after intravitreal injection.[299]NanoparticlesPLGApDNASignificant reduction of vascular leakage CNV induced diabetic rats without tissue damage and effective K5 expression in the retinal layer for up to 4 weeks after intravitreal injection.[300]Nano-ballsbPEIsiRNASustained release, longer retention with effective concentration in choroid and RPE target tissues for up to 2 weeks after intravitreal injection.[301]NanoparticlesDOTAP, DOPEpDNATransfection effect is increased significantly up to 6-fold in RPE as well as 2-fold in in vitro after intravitreal injection. [302]Lipid-nanoparticlesDOTAP, Cholesterol, PEG-DSPE siRNAVEGFR1 expression inhibited in ARPE-19 cell-lines showing no tissue damage and reduced CNV areas after intravitreal injection.[303]SLNPrecirol (ATO5), DOTAP, Tween 80 Dextran, HApDNAHigh transfection efficiency in both PR and INL showing improvement in retinal structure after 2 weeks after intravitreal injection.[304]NLCMonolaurin, Monostearin, Glyceryl tripalmitate, Palmitin Glyceryl stearateSorafenibDemonstrated excellent physicochemical properties and good tolerance, sustained release, and enhanced ocular bioavailability in CNV after topical administration.[305]NLCGlyceryl monostearate, lipoplysaccharidesDasatinibObserved sustained release, reduced ocular toxicity, and facilitated penetration into cornea via topical administration with effective inhibition of CNV.[306]DendrimerLecithinAnti-VEGF Plus Oligonucleotide 1Topical delivery of dendrimers plus ODN-1 to the eyes of rats and inhibited laser-induced CNV for up to 6 months.[307]NiosomesDOTAP, squalene, Polysorbate 80pDNAPersistent protein expression after transfection intravitreally for at least 1 month after injection. Protection against enzymatic digestion, providing broad surface transfection in inner layer of retina with no cytotoxicity.[224]LiposomesDOTMA, Cholesterol, DOPEpDNAHighest transfection effect with lower quantity of plasmid DNA in liposomes reached peak level within 3 days after intravitreal injection.[228]LiposomesPC, Cholesterol, PEG-DSPEOligonucleotideSustained release in vitreous and retina–choroid from intravitreal injection of liposomal oligonucleotides showing protective effect against enzymatic degradation after intravitreal injection.[229]LiposomesDPPC, EPC, CholesterolBevacizumabSignificant improvement in mean residence time of bevacizumab after intravitreal injection.[308]LiposomesEPC-Chol and DPC-CholesterolBevacizumabShowed sustained release of drug for up to 42 days after intravitreal injection.[221]LiposomesPC-PS(Cholesterol)TocBevacizumab-(annexin)Liposomes showed 100 nm of size prepared with dehydration–rehydration technique and coated with annexin after intravitreal injection.[220]LiposomesPC, CholesterolVasoactive intestinal peptideReduced inflammation significantly with prolonged protection of peptide up to 14 days in vivo after intravitreal injection.[225]Biodegradable ImplantsMolded hydrogel matrixBevacizumabImplants prepared using PRINT technology from molded hydrogel showed sustained release of bevacizumab for 2 months after intravitreal administration. [159]Biodegradable implantsPCLRanibizumabFilm device containing nanopores on PCL provided sustained release of ranibizumab for 3 months after intravitreal injection. [309]Nonbiodegradable implantsProgrammable micropump deviceRanibizumabMicropump for posterior segment prepared using nonbiodegradable polymers showed long-term release of ranibizumab after intravitreal administration.[310]ImplantsPort delivery systemRanibizumabSemipermeable refillable membrane providing long-term release of ranibizumab for 1 year after intravitreal administration.[311]Non- biodegradable implantsNT-503VEGFR-FcIncreased specific VEGFR binding observed with encapsulating cell showing continuous production of therapeutic proteins for two years after intravitreal administration. [312]Verisome IB20089Biodegradable implant with liquid gelTriamcinolone/RanibizumabThe formed spherules provided long term sustained release up to 1 year after intravitreal injection. [313]Microsphere in hydrogelPNIPAAmRanibizumab and afliberceptSustained in vitro release for up to 196 days from thermosensitive hydrogel after intravitreal injection.[180]In situ hydrogelHA(DEX)BevacizumabHydrogel formed by chemical crosslinking showed sustained release for up to 6 months in vivo administered intravitreal in rabbit model.[146]HydrogelAlginate(Chitosan) hydrogel/PLGA microspheresBevacizumab/RanibizumabSustained release from intravitreally administered hydrogel observed for both bevacizumab and ranibizumab for up to 3 months. [158]HydrogelPLGA-mPEGBevacizumabSustained release for up to 1 month via intravitreal route in rabbits.[159]HydrogelPEOz(PCL)PEOzBevacizumabSustained release for up to 20 days in vitro. [160]Silk based hydrogelsSilk fibroin BevacizumabSustained release of bevacizumab from intravitreally administered hydrogel for 90 days in vitro as well as in vivo in Dutch-belted rabbits.[161]In situ gelPCM(HEMA)BevacizumabProvided in vivo retention for up to 2 months in SD rats via suprachoroidal route.[162]HydrogelPNIPAAm InsulinSustained release of insulin for up to 30 days in vitro. [163]HydrogelPLGA–PEGOvalbumin (model protein)Provided sufficient protein concentration for up to 14 days in ocular tissues via subconjunctival route.[164]HydrogelESHUBevacizumabSustained release for up to 9 weeks via intravitreal route in rabbits.[166]HydrogelPEG-(ESHU)BevacizumabShowed good in vitro and in vivo biocompatibility after intravitreal injection.[167]HydrogelPNIPAAmBevacizumab/RanibizumabProvided good mechanical properties, biocompatibility from thermosensitive hydrogel after intravitreal injection. [170]Diels-alder hydrogelsPEG MacromonomersBevacizumabProvided mechanical stability and long-term release for up to 6 weeks from the chemically crosslinked hydrogel after intravitreal injection. [314]Encapsulated cellsPolysulfoneCNTFContinued clinical trials.[315]Retinal cellsHASConnexin43 mimetic peptideSustained release and prolonged retention with suppression of RGC and inflammation observed when intravitreally injected NPs encapsulating Cx43 MP were evaluated in a rat model of retinal ischemia-reperfusion injury.[316]Abbreviations: bPEI: Branched polyethylenimine; CNV: corneal neovascularization; CNTF: ciliary neurotrophic factor; DA: dexamethasone acetate; DDS: drug delivery systems; DEX: dexamethasone; DOPC: 1,2-Dioleoyl-sn-glycero-3-phosphocholine; DOPE: 1,2-Dioleoyl-sn-glycero-3-phospho-ethanol-amine; DOPG: 1,2-Dioleoyl-sn-glycero-3-phospho (10-rac-glycerol); DOTAP: 1,2-Dioleoyl-3-trimethyl-ammonium-propane; DOTMA: 1,2-Di-o-octadecanyl-3-tri-methyl-ammonium-propane; DPPC: 1,2-Dipalmitoyl-sn-glycero-3-phosphocholine; DR: diabetic retinopathy; DSPE (PEG): 1,2-Distearoyl-SN-Glycero-3-phosphoethanolamine-n(amino-polyethylene glycol); EAU: experimental autoimmune uveitis; EIU: endotoxin-induced uveitis; EPC: egg phosphatidylcholine; HA: hyaluronic acid; HAS: human serum albumin; HSPC: 1 (a)-Phosphatidyl-Choline-Hydrogenated soy; INL: inner nuclear layer; IPL: inner plexiform layer; mPEG (PCL): monomethoxy poly-ethylene glycol-poly-e-caprolactone; NDPR: nonproliferative diabetic retinopathy; NPs: nanoparticles; PCL: polycaprolactone; PC-PS-Toc: egg phosphatidylcholine-porcine brain phosphatidylserine-tocopherol; pDNA: plasmid deoxy-ribonucleic acid; PEG: polyethylene glycol; PEG-PHDC: poly(methoxy-poly(ethylene glycol)-cyanoacrylate-co-hexadecyl cyanoacrylate); PEO-PCL-PEO: poly(2-ethy-2-oxazoline)b-poly(caprolactone)-b-poly(2-ethyl-2-oxazoline); PLA: polylactic acid; PLA–PEO: poly lactic acid-polyethylene oxide; PLGA: poly-(lactide-coglycolide); PNIPAAm: poly (n-isopropylacrylamide); PSHU: poly (serinol hexamethylene urethane); RGC: retinal ganglion Cell; RPE: retinal pigment epithelium; RVO: retinal vein occlusion; SDS: Prague-Dawley; siRNA: small interfering ribonucleic acid; SLN: solid lipid nanoparticles; STZ: streptozotocin; VEGF: vascular endothelial growth factor; VIP: vasoactive intestinal peptide.

## 6. Conclusions

Recent developments in the medical biotechnology area promoted the use of therapeutic proteins to treat different ocular diseases and have changed the scenario in the research work carried out in last few decades. Optimal efficacy can be achieved with proper knowledge of ocular barriers, nature, and pharmacokinetics of therapeutic proteins along with reduction of the dosing frequency and use of novel or combination of technologies (e.g., nanocarriers included in hydrogel-based systems) or by adding penetration enhancers or enzyme inhibitors to the formulations. Future research must be conducted toward the development of more efficient, stable, noninvasive, and cost-effective formulations for ocular delivery of therapeutic proteins.

The development of effective delivery systems containing stable therapeutic proteins that can reach the eye topically, subconjunctivally, or periocularly remains challenging. There is a need to establish pharmacokinetic models that provide useful insights into the development of these ocular delivery systems, which can aid in preclinical to clinical translation and in predicting the dosing regimen.

## Figures and Tables

**Figure 1 pharmaceutics-15-00205-f001:**
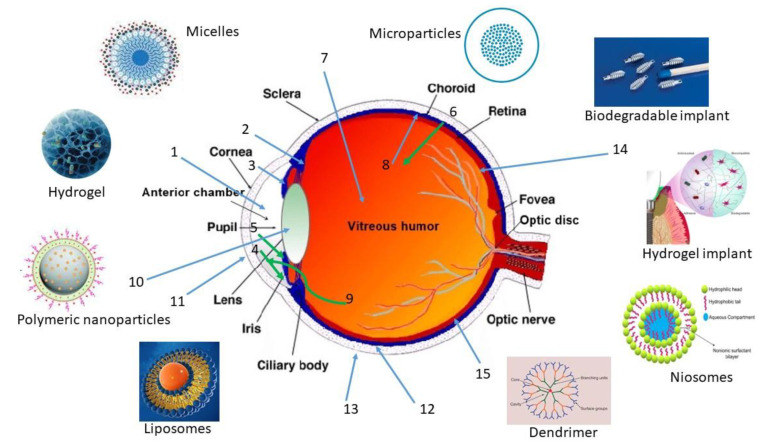
Schematic representation of various formulation approaches and routes of administration to the ocular tissues. 1. Transcorneal permeation into the anterior chamber, 2. Noncorneal drug permeation across conjunctiva to sclera into anterior uvea, 3. Drug distribution into anterior chamber from blood stream through blood aqueous barrier, 4. Drug elimination from anterior chamber by aqueous humor to trabecular meshwork and Schlemm’s canal, 5. Elimination of drug from aqueous humor into systemic circulation across blood aqueous barrier, 6. Distribution of drug from blood into posterior segment across the blood retinal barrier, 7. Intravitreal route, 8. Drug elimination from vitreous via posterior route across blood retinal barrier, 9. Elimination of drug from vitreous via anterior route to posterior chamber, 10. Intracameral route, 11. Intrastromal route, 12. Subconjunctival route, 13. Subtenon route, 14. Suprachoroidal route, 15. Subretinal route.

**Figure 2 pharmaceutics-15-00205-f002:**
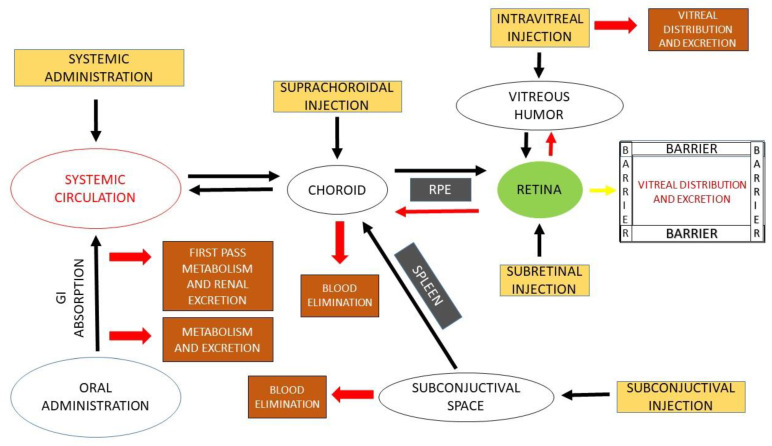
Different routes for ocular drug clearance/elimination.

**Figure 3 pharmaceutics-15-00205-f003:**
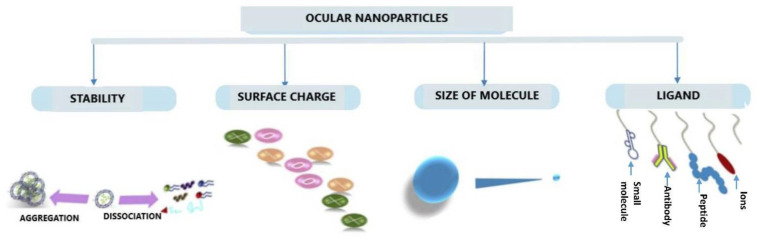
Characteristics of the ocular nanoparticles that affect their intraocular distribution and elimination.

**Figure 4 pharmaceutics-15-00205-f004:**
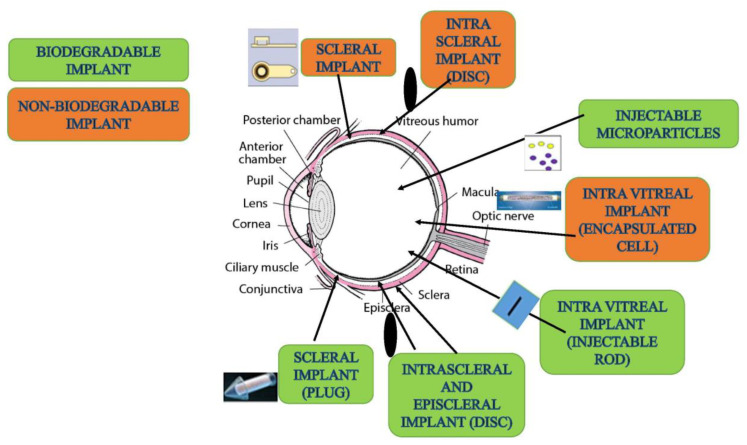
Biodegradable and nonbiodegradable implants for ocular delivery of therapeutic proteins.

**Table 2 pharmaceutics-15-00205-t002:** Examples of different approaches used to improve the bioavailability of ocular therapeutic proteins.

Approaches	Remarks
Use of penetration enhancers	Increases corneal permeability.
Improve enzymatic resistance	Avoids degradation by enzymes present in ocular tissues.
**Conjugation Approaches**	
Conjugation with ligands	Tissue-specific delivery with minimal toxicity and minimal systemic exposures.
Conjugation with ligand, lipids, melanin, hyaluronan, and PEG	Improves half-lives or reduces immunogenicity, protects macromolecules, prevents proteolytic degradation, and provides long-term stability/stability. Improves membrane penetration.
**Formulation Approaches**	
Hydrogels	Protect molecules from degradation and provide long-term release.
Micro/Nanocarriers	Protect molecules from enzymatic degradation. Enhance permeation. Restrict drug release to the desired area of eye. Provide depot and prolonged retention of the formulation. Improve physical stability. Increase drug permeability.
Mucoadhesive polymeric system	Achieve site-specific drug delivery. Improves permeation.
Cell-Penetrating Peptides	Improve penetration overcoming ocular barriers and provide sustained release by preventing degradation and increasing the drug residence time.
Encapsulated Cell Technology	Enzyme protection, stability, long-term release.
Iontophoresis	Improves permeation, provides transfer of ionized molecules through biological membranes using low electric current to the ocular tissues.
Microneedles	Provide passive diffusion of therapeutics, overcoming the transport barriers of epithelial tissues, eliminating clearance by conjunctival mechanisms, and minimizing retinal damage.
Implants	Protect from degradation and provide depot with slow and sustained release at constant rate over extended period.

## Data Availability

Not applicable.

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
