# Peer review of "Ocular Delivery of Therapeutic Proteins: A Review"

_pharmaceutics, 2023, doi:10.3390/pharmaceutics15010205_

Round 1

Reviewer 1 Report (New Reviewer)

The authors presented a comprehensive review of the delivery of therapeutic proteins to ocular tissues, from the aspects of ocular drug administration routes, ocular barriers and administration approaches to conjugation and formulation approaches. Detailed comments are as below:

In abstract, line 22, it should be ‘age-related macular degeneration’.

In introduction, line 44, age-related macular degeneration is not a complication of DR. It is a disease entity by itself.

In Table 1, please add a column summarizing the target of each molecule, for example, VEGF, PDGF and etc.

Section 3 and section 3.1 titles, lines 245 and 246: OCULAR was misspelled.

In section 4 conjugation approaches, it should also mention table 2 since table 2 contains some content of conjugation approaches.

Section 5 title, line 554: APPROACHES was misspelled.

Author Response

Reviewer 2 Report (New Reviewer)

This is an extensive review of the advances in the ocular release of proteins. This is a complete review that may be of interest to those researchers who are beginning to investigate the ocular administration of proteins. The review is written in understandable language. The initial part of the review is very general and hardly any new information compared to other reviews published in Pharmaceutics. In addition, although it handles a large number of publications, it does not cite current works on the administration of proteins such as lactoferrin for the treatment of corneal neurodegeneration, among others. The most interesting part is the one that focuses on the promotion of absorption and the preparation of the release systems. Still, I think it's interesting I consider that it can be published in Pharmaceutics

Author Response

Reviewer 3 Report (New Reviewer)

The review  reports a very wide collection of  references about the formulative approaches for the delivery  of proteins in ocular field  focusing mainly  on the treatment of ocular posterior chamber  pathologies.   This item could be highlighted in the title,  although both in the abstract  (lines 33-34) and in the introduction  (lines 112-114) the authors refer to   drug delivery in anterior and posterior chambers. 

Some parts of the manuscript should be refined to improve clarity and to balance the coverage of the different methodologies used for ocular delivery.

The revision should essentially concern: - the reduction of  paragraph n.2 with a  reduction of  the different routes for the delivery of therapeutic proteins (Lines 124-138);  - the introduction of some references in paragraph n . 4; - an accurate order in the number of the references; and 

 - some other references should be added  to   improve the knowledge on specific formulative techniques  for the  ocular delivery of therapeutically active proteins.

In detail:

- the reference [35] does not seem appropriate since refers mainly on oral administration of peptides; references [36] is not seem reported  in the test; the order of references [37] is wrong and  it has to slip to after the [53]; the references from [85] to [99]  are not reported in the right order along the manuscript  and they should be revised;

- the  suggested references are:

GadziÅ„ski et al.,  Beilstein J. Nanotechnol. 2022, 13, 1167–1184.

https://doi.org/10.3762/bjnano.13.98

Bhattacharya et al., Journal of Controlled Release 327 (2020) 584–594.

https://doi.org/10.1016/j.jconrel.2020.09.005

Burgalassi et al., Drug Deliv Transl Res. 2018 Jun;8(3):461-472.

doi: 10.1007/s13346-018-0520-x

Ilochonwu et al., Biomacromolecules 2022, 23, 2914−2929.

https://doi.org/10.1021/acs.biomac.2c00383

Author Response

This manuscript is a resubmission of an earlier submission. The following is a list of the peer review reports and author responses from that submission.

Round 1

Reviewer 1 Report

The review did a very through summary of ocular delivery of therapeutic protein. It provided some insight to the protein drug delivery to the eye. Below are some comments:

1) Overall, the literatures used in the review are not that up to date. Literatures within 3 years take up about 10% of all the literatures. 

2) It would be better for the authors to use levels of titles to define different sections and considerations. It is overwhelming to read a whole section without any subtitles and number of levels. 

3) Lines 192-193, please make the sentence in the right format.

4) Table 1: The review is supposed to discuss all the protein drug. However, dexamethasone is also listed. In addition, the molecular weight is not 150 (kD). Please remove it.

5) Figure 3: In the ligand part, the elements should be labeled. Otherwise, it will cause confusion.

6) Line 597: typo, cytoxicity should be cytotoxicity.

7) Lines 600-605, Here the example should demonstrate the low toxicity. Instead, the authors seemed to put the bioavailability data. Although higher bioavailability does not necessarily mean low toxicity, it would be better to show the direct data of toxicity such as survival test. 

8) Line 918 This paragraph is definitely a separate section regarding sterilization. It should be titled with a different section. 

Author Response

Respected Sir,

All the authors are highly thankful to the time and efforts taken by the expert reviewer and giving their valuable suggestions in the manuscript. I have made the corrections as per the suggestions in the manuscript. Please find the Answers to the Comments in the attachment.

Thanking you once again.

Regards,

Author.

Reviewer 2 Report

The review manuscript entitled "Ocular Delivery of Therapeutic Proteins: Recent Perspectives" discuss interesting data about the limitations of the administration routes and different strategies to overcome the current challenges for ocular therapies, in particular delivering therapeutic proteins.

Although this study provides some interesting information in this topic, the lack of organization and use of English greatly difficult the comprehension and the understanding of the main messages that authors wish to transmit to the readers. In my opinion, I cannot recommend this review for further consideration and publication.

However, I would like to suggest some issues to authors, that may be would like to take in consideration, in order to improve some basic aspects of the review.

Main suggestions for the authors are:

- Include headings and subheadings to organize the content in a comprehensive way. Only 1.Introduction and 2.Conclusions is insufficient. Unfortunately, as it is, is not possible to understand the main topics that authors wish to discuss and why.

- Present figures as they are named in the text. It is confusing to find first in the text Figure 2 instead of figure 1.

- Revise the text to avoid duplicated text, (for example, lines 597-600 are repeated in lines 606-609) and contradictions (for example, line 436 Conjugation with lipid derivatives section discuss about alternative strategies for not using lipid derivatives).

- Text format: separate the numbers and units throughout the text; describe abbreviations the first time they are mentioned in the text.

- References: please revise all the bibliography since some ref. do not correspond with the issues discussed in the text (for example, line 739 discuss discosomes but the reference 180 corresponds to nanostructured lipid carriers; line 745 is about liposomes but the reference 184 corresponds to magnetic nanoparticles, ...). I would also suggest to revise the format of the references in the text (for example, line 722 [177-178] should be [177, 178] and order of appearance in accordance to the text/tables.

From my point of view, Table 2 best resemble the main topic of the review. May be authors could consider to add another column to indicate the route of administration of each approach, since this topic seems to be also of importance in the authors discussion throughout the review.

Author Response

(The authors gave the same response as above.)

Reviewer 3 Report

The authors review literature on ocular delivery of protein based therapeutics with emphasis on drug delivery systems in the ocular space. Overall, the topic of the review is of reasonable interest. I have the below comments for the authors.

1.     The structure of the manuscript needs to be better organized. Several sub-sections are lumped as paragraphs within introduction, which seems odd. The text should be structured into additional main sections beyond introduction and conclusions (e.g., Routes of administration, Ocular PK, drug delivery systems etc). Having a clear structure with numbered sections and sub-sections will improve the review and help guide the reader (e.g., Section 1; Sub-sections 1.1, 1.2, 1.3 and so on).

2.     Also, suggest improving the structure of the section “BARRIERS AND ROUTES OF ADMNISTRATION”. Recommend something along the lines of having a brief intro para and including 2 broad sub-sections (1) extra-ocular RoA (combine current BARRIERS AND ROUTES OF ADMNISTRATION section and systemic route section) (2) ocular RoA (combining sections on topical, intraocular, periocular etc)

3.     Some sections are too long to read and follow.

a.     “OCULAR PHARMACOKINETICS & APPROACHES:” section is loo lengthy as written and breaking it down into sub-sections will be useful.

b.     “Injectable nanocarriers:” - Again, suggest breaking this down in to sub-sections to improve readability.

4.     Below mentioned is stated in the review but here is no mention of target-mediated drug distribution and its role on influencing the PK of therapeutic agents. Suggest including a breif discussion in this regard.

“Ocular distribution of protein therapeutics to the eye depends on several factors, 233 such as membrane permeability, ocular elimination, non-target binding, and degradation 234 by proteolytic enzymes.”

5.     Suggest the following changes for the tables.

a.     Table 1: The kD for Farcimab is not listed. Suggest including it or stating the reason for not including it.

b.     Table 2: Including another column stating which test species was used in the listed studies would be informative (e.g., Preclinical species / clinical testing / in vitro based etc). Without this information, the table offers limited value.

6.     I have the following comments for the figures.

a.     Fig 1: The text describing the processes shown by numbers should be part of the figure legend.

b.     Fig 2: Suggest showing aqueous compartment for completeness. Clearance from vitreous >> aqueous >> blood is not shown. Also suggest including a brief description of the schematic in the figure legend in addition to the figure title.

c.     Fig 3: The figure seems to be over simplified and does not add any value. Suggest including more details in the figure to make is useful or removing it.

7.     Including a schematic figure to summarize the classification within ocular drug delivery systems that are discussed (invasive: micro/nano carriers, implants, hydrogels etc; non-invasive: CPP, iontophoresis etc) would be useful. Also restructuring the relevant sections to match this classification would improve flow and readability.

a.     For instance, “INJECTABLE HYDROGELS” section seems out of place and should be discussed before “NON-INVASIVE TECHNIQUES:” section.

8.     This article focuses more on review of literature and so suggest removing ‘recent perspectives’ from the title.

9.     Several statements in the review are unclear and need to be rephrased for clarity. Below stated are a few examples.

“Macula gets 48 thinner (atrophic) with age in some patients known as dry AMD, while advanced neo-49 vascular AMD known as wet AMD or CNV [3]. New vessels growth is an important 50 cause for central RVO with abrupt onset, leading to capillary occlusion inducing hypoxia 51 increasing vascular endothelial growth factor (VEGF) expression and consequently 52 ended in retinal proliferation of new vessels.”

“Buccal, intranasal and inhalation routes present extremely low bioavailabil-115 ity with bypass hepatic and GI metabolism but are less invasive and show better patient 116 compliance. There are three intranasal peptides approved (desmopressin, nafarelin and 117 calcitonin) that show 3% bioavailability [12-13].”

“Large molecular drugs bypass the 166 corneal epithelium penetration route and will goes to non-corneal absorption makes the 167 potential for delivery of large molecular proteins and peptide delivery, which showed 168 poor corneal permeability [18].”

“From the posterior 190 segment diffused to ILM and finally reaches to retina (Figure 2).”

“Moreover, due to high blood flow in choroidal blood vessels most of the amount of drug 211 lost to systemic circulation lead to poor half-lives.”

“As excellent membrane modulating capacity, cell penetrating peptides are exten-791 sively used as penetration enhancers to overcome the barriers of ocular tight junctions 792 and provide effective delivery of drugs to the ocular tissues. They are natural, synthetic amino acid sequences containing peptides, which facilitate the hydrophilic cargo of pep-794 tide, protein, and genes into the intracellular ocular tissues [200]. Involving non-invasive 795 or less invasive treatment and with the capability to cross through the biological mem-796 branes, the use of cell penetrating peptides for ocular delivery of drugs is being paid 797 much more attention.”

10.  Please address the following.

·      Line 101: The below sentence says interior, should it be anterior instead?

ocular tissues both interior and 101 posterior segments of the eye of therapeutic proteins

·      Line 191: define ILM before using it as an abbreviation.

·      Line 216: The intent of below stated is unclear. Rapid clearance and long retention times contradict each other.

Though 216 rapid clearance, the particles show long retention time that can last up to months.”

·      Line 228: The below is stated in the periocular section. Shouldn't this be less invasive? The first line of the paragraph also says that it is less invasive.

This route is more 228 invasive and...”

11.  The below mentioned is stated in the Formulation approaches section. However, this text is similar to the text stated in "periocular route" section and so is repetitive. 

“Periocular route, a non-invasive alternative to intravitreal route where the macro-480 molecules are introduced at the subconjunctival, subtenon, retrobulbar or posterior jux-481 tascleral spaces with or without the use of carrier without damage to retina or eyeball 482 providing drug directly to the target tissues like retina or retinal pigment epithelium [38]. 483 Subconjunctival delivery is another less invasive and alternate route to subretinal, su-484 prachoroidal or subtenon route of delivery see Figure 1.”

12.  Please check for typos, grammar and language throughout the review.

Author Response

(The authors gave the same response as above.)

Round 2

Reviewer 2 Report

The new version of the review manuscript entitled "Ocular Delivery of Therapeutic Proteins: A Review" is now more organized in terms of sections, which makes more easy to understand the focus of the work. However, overall, there is a lack of consonance throughout the manuscript, with mistakes (English language, references and figures in text not matching) and misconceptions that, from my humble point of view, impede that this manuscript is accepted for publication.

In this new version, the authors have made only minor corrections, focusing only on changing the examples of corrections exposed in the first peer review report, without revising these issues in the entire manuscript.

The naming of the figures in the text and illustrations do not match, also there is a huge amount of references in the reference list that do not match with the studies explained throughout the manuscript. In the tables, which are excepted to summarize experimental studies in the field, appear many reviews in the references column. Table 1 legend introduce some kind of comparison "Comparison of the molecular characteristics of anti-vascular endothelial growth factor 312 (VEGF) antibodies with other anti-VEGF agents", but there is no any comparison. Units and quantities are still without space in many cases.

In figure 1, the subretinal injection is lacking, while intravitreal injection is repeated (7 and 15).

Figure 2 is difficult to understand. How can dermal administration appear in a figure intended for ophthalmic drugs?

In figure 3 the main factor that determines the stability, zeta potential and size of a nanorparticle is the composition of the nanoparticle itself (lipidic, polymeric, ....). In addition,  these parameters do not affect only at the intravitreal level but also topically, subretinally,....

Figure 5 makes no sense from my point of view. Why micro/nanocarriers are considered an invasive technique? they have also been used for ocular surface therapies. What about the cell encapsulation strategy? why it does not appear in the scheme?

Again, point 3.2.1. is intended for explaining the "conjugation with lipid derivates" but it explains the alternative strategies to do not use these lipid derivatives. It makes no sense.

In general there is text that does not match with the section in which it is described. Why are intravitreal studies explained in the section intended for non-invasive techniques? Intravitreal injections are invasive approaches. As I am concern, microneedles are not a type of iontophoresis technique.

These and many other issues found in the peer review process seem to me enough for not accepting this manuscript for publication.